# Diffusion for World Modeling:
# Visual Details Matter in Atari[†]

**Eloi Alonso**[*]
University of Geneva

**Adam Jelley**[*]
University of Edinburgh

**Vincent Micheli**
University of Geneva

**Anssi Kanervisto**
Microsoft Research

**Amos Storkey**
University of Edinburgh

**Tim Pearce**[‡]
Microsoft Research

**François Fleuret**[‡]
University of Geneva

## Abstract

World models constitute a promising approach for training reinforcement learning agents in a safe and sample-efficient manner. Recent world models predominantly operate on sequences of discrete latent variables to model environment dynamics. However, this compression into a compact discrete representation may ignore visual details that are important for reinforcement learning. Concurrently, diffusion models have become a dominant approach for image generation, challenging well-established methods modeling discrete latents. Motivated by this paradigm shift, we introduce DIAMOND (DIffusion As a Model Of eNvironment Dreams), a reinforcement learning agent trained in a diffusion world model. We analyze the key design choices that are required to make diffusion suitable for world modeling, and demonstrate how improved visual details can lead to improved agent performance. DIAMOND achieves a mean human normalized score of 1.46 on the competitive Atari 100k benchmark; a new best for agents trained entirely within a world model. We further demonstrate that DIAMOND's diffusion world model can stand alone as an interactive neural game engine by training on static *Counter-Strike: Global Offensive* gameplay. To foster future research on diffusion for world modeling, we release our code, agents, videos and playable world models at `https://diamond-wm.github.io`.

## 1  Introduction

Generative models of environments, or "world models" (Ha and Schmidhuber, 2018), are becoming increasingly important as a component for generalist agents to plan and reason about their environment (LeCun, 2022). Reinforcement Learning (RL) has demonstrated a wide variety of successes in recent years (Silver et al., 2016; Degrave et al., 2022; Ouyang et al., 2022), but is well-known to be sample inefficient, which limits real-world applications. World models have shown promise for training reinforcement learning agents across diverse environments (Hafner et al., 2023; Schrittwieser et al., 2020), with greatly improved sample-efficiency (Ye et al., 2021), which can enable learning from experience in the real world (Wu et al., 2023).

Recent world modeling methods (Hafner et al., 2021; Micheli et al., 2023; Robine et al., 2023; Hafner et al., 2023; Zhang et al., 2023) often model environment dynamics as a sequence of discrete latent variables. Discretization of the latent space helps to avoid compounding error over multi-step time horizons. However, this encoding may lose information, resulting in a loss of generality and reconstruction quality. This may be problematic for more real-world scenarios where the information

---

[†]To prevent confusion, this is the final version of (Alonso et al., 2023) and is not related to (Ding et al., 2024).
[*]Equal contribution. [‡]Equal supervision. Contact: `eloi.alonso@unige.ch` and `adam.jelley@ed.ac.uk`

38th Conference on Neural Information Processing Systems (NeurIPS 2024).

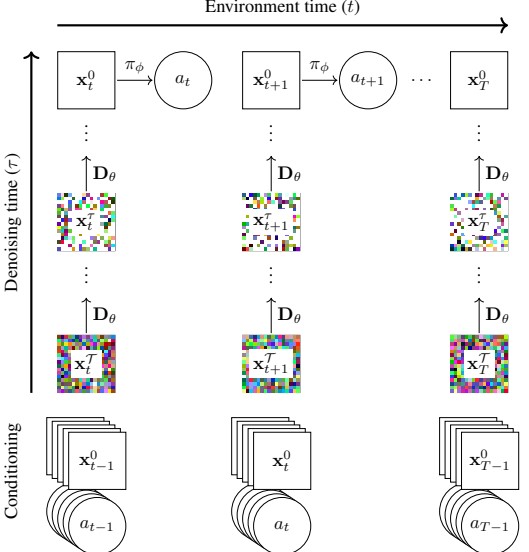

Figure 1: Unrolling imagination of DIAMOND over time. The top row depicts a policy $\pi_\phi$ taking a sequence of actions in the imagination of our learned diffusion world model $\mathbf{D}_\theta$. The environment time $t$ flows along the horizontal axis, while the vertical axis represents the denoising time $\tau$ flowing backward from $\mathcal{T}$ to $0$. Concretely, given (clean) past observations $\mathbf{x}_{<t}^0$, actions $a_{<t}$, and starting from an initial noisy sample $\mathbf{x}_t^{\mathcal{T}}$, we simulate a reverse noising process $\{\mathbf{x}_t^\tau\}_{\tau=\mathcal{T}}^{\tau=0}$ by repeatedly calling $\mathbf{D}_\theta$, and obtain the (clean) next observation $\mathbf{x}_t^0$. The imagination procedure is autoregressive in that the predicted observation $\mathbf{x}_t^0$ and the action $a_t$ taken by the policy become part of the conditioning for the next time step. Animated visualizations of this procedure can be found at `https://diamond-wm.github.io`.

required for the task is less well-defined, such as training autonomous vehicles (Hu et al., 2023). In this case, small details in the visual input, such as a traffic light or a pedestrian in the distance, may change the policy of an agent. Increasing the number of discrete latents can mitigate this lossy compression, but comes with an increased computational cost (Micheli et al., 2023).

In the meantime, diffusion models (Sohl-Dickstein et al., 2015; Ho et al., 2020; Song et al., 2020) have become a dominant paradigm for high-resolution image generation (Rombach et al., 2022; Podell et al., 2023). This class of methods, in which the model learns to reverse a noising process, challenges well-established approaches modeling discrete tokens (Esser et al., 2021; Ramesh et al., 2021; Chang et al., 2023), and thereby offers a promising alternative to alleviate the need for discretization in world modeling. Additionally, diffusion models are known to be easily conditionable and to flexibly model complex multi-modal distributions without mode collapse. These properties are instrumental to world modeling, since adherence to conditioning should allow a world model to reflect an agent's actions more closely, resulting in more reliable credit assignment, and modeling multi-modal distributions should provide greater diversity of training scenarios for an agent.

Motivated by these characteristics, we propose DIAMOND (DIffusion As a Model Of eNvironment Dreams), a reinforcement learning agent trained in a diffusion world model. Careful design choices are necessary to ensure our diffusion world model is efficient and stable over long-time horizons, and we provide a qualitative analysis to illustrate their importance. DIAMOND achieves a mean human normalized score of 1.46 on the well-established Atari 100k benchmark; a new state of the art for agents trained entirely within a world model. Additionally, operating in image space has the benefit of enabling our diffusion world model to be a drop-in substitute for the environment, which provides greater insight into world model and agent behaviors. In particular, we find the improved performance in some games follows from better modeling of critical visual details. To further demonstrate the effectiveness of our world model in isolation, we train DIAMOND's diffusion world model on 87 hours of static *Counter-Strike: Global Offensive* (CSGO) gameplay (Pearce and Zhu, 2022) to produce an interactive neural game engine for the popular in-game map, *Dust II*. We release our code, agents and playable world models at `https://diamond-wm.github.io`.

## 2 Preliminaries

### 2.1 Reinforcement learning and world models

We model the environment as a standard Partially Observable Markov Decision Process (POMDP) (Sutton and Barto, 2018), $(\mathcal{S}, \mathcal{A}, \mathcal{O}, T, R, O, \gamma)$, where $\mathcal{S}$ is a set of states, $\mathcal{A}$ is a set of discrete actions, and $\mathcal{O}$ is a set of image observations. The transition function $T : \mathcal{S} \times \mathcal{A} \times \mathcal{S} \to [0, 1]$ describes the environment dynamics $p(\mathbf{s}_{t+1} \mid \mathbf{s}_t, \mathbf{a}_t)$, and the reward function $R : \mathcal{S} \times \mathcal{A} \times \mathcal{S} \to \mathbb{R}$ maps transitions to scalar rewards. Agents cannot directly access states $s_t$ and only see the environment through image observations $x_t \in \mathcal{O}$, emitted according to observation probabilities $p(\mathbf{x}_t \mid \mathbf{s}_t)$,

described by the observation function $O : \mathcal{S} \times \mathcal{O} \to [0, 1]$. The goal is to obtain a policy $\pi$ that maps observations to actions in order to maximize the expected discounted return $\mathbb{E}_\pi[\sum_{t \geq 0} \gamma^t r_t]$, where $\gamma \in [0, 1]$ is a discount factor. World models (Ha and Schmidhuber, 2018) are generative models of environments, i.e. models of $p(s_{t+1}, r_t \mid s_t, a_t)$. These models can be used as simulated environments to train RL agents (Sutton, 1991) in a sample-efficient manner (Wu et al., 2023). In this paradigm, the training procedure typically consists of cycling through the three following steps: collect data with the RL agent in the real environment; train the world model on all the collected data; train the RL agent in the world model environment (commonly referred to as "in imagination").

## 2.2 Score-based diffusion models

Diffusion models (Sohl-Dickstein et al., 2015) are a class of generative models inspired by non-equilibrium thermodynamics that generate samples by reversing a noising process.

We consider a diffusion process $\{\mathbf{x}^\tau\}_{\tau \in [0, \mathcal{T}]}$ indexed by a continuous time variable $\tau \in [0, \mathcal{T}]$, with corresponding marginals $\{p^\tau\}_{\tau \in [0, \mathcal{T}]}$, and boundary conditions $p^0 = p^{data}$ and $p^{\mathcal{T}} = p^{prior}$, where $p^{prior}$ is a tractable unstructured prior distribution, such as a Gaussian. Note that we use $\tau$ and superscript for the diffusion process time, in order to keep $t$ and subscript for the environment time.

This diffusion process can be described as the solution to a standard stochastic differential equation (SDE) (Song et al., 2020),

$$d\mathbf{x} = \mathbf{f}(\mathbf{x}, \tau)d\tau + g(\tau)d\mathbf{w}, \tag{1}$$

where $\mathbf{w}$ is the Wiener process (Brownian motion), $\mathbf{f}$ a vector-valued function acting as a drift coefficient, and $g$ a scalar-valued function known as the diffusion coefficient of the process.

To obtain a generative model, which maps from noise to data, we must reverse this process. Remarkably, Anderson (1982) shows that the reverse process is also a diffusion process, running backwards in time, and described by the following SDE,

$$d\mathbf{x} = [\mathbf{f}(\mathbf{x}, \tau) - g(\tau)^2 \nabla_\mathbf{x} \log p^\tau(\mathbf{x})]d\tau + g(\tau)d\bar{\mathbf{w}}, \tag{2}$$

where $\bar{\mathbf{w}}$ is the reverse-time Wiener process, and $\nabla_\mathbf{x} \log p^\tau(\mathbf{x})$ is the (Stein) score function, the gradient of the log-marginals with respect to the support. Therefore, to reverse the forward noising process, we are left to define the functions $f$ and $g$ (in Section 3.1), and to estimate the unknown score functions $\nabla_\mathbf{x} \log p^\tau(\mathbf{x})$, associated with marginals $\{p^\tau\}_{\tau \in [0, \mathcal{T}]}$ along the process. In practice, it is possible to use a single time-dependent score model $\mathbf{S}_\theta(\mathbf{x}, \tau)$ to estimate these score functions (Song et al., 2020).

At any point in time, estimating the score function is not trivial since we do not have access to the true score function. Fortunately, Hyvärinen (2005) introduces the *score matching* objective, which surprisingly enables training a score model from data samples without the knowledge of the underlying score function. To access samples from the marginal $p^\tau$, we need to simulate the forward process from time 0 to time $\tau$, as we only have clean data samples. This is costly in general, but if $f$ is affine, we can analytically reach any time $\tau$ in the forward process in a single step, by applying a Gaussian perturbation kernel $p^{0\tau}$ to clean data samples (Song et al., 2020). Since the kernel is differentiable, score matching simplifies to a *denoising score matching* objective (Vincent, 2011),

$$\mathcal{L}(\theta) = \mathbb{E}\left[\|\mathbf{S}_\theta(\mathbf{x}^\tau, \tau) - \nabla_{\mathbf{x}^\tau} \log p^{0\tau}(\mathbf{x}^\tau \mid \mathbf{x}^0)\|^2\right], \tag{3}$$

where the expectation is over diffusion time $\tau$, noised sample $\mathbf{x}^\tau \sim p^{0\tau}(\mathbf{x}^\tau \mid \mathbf{x}^0)$, obtained by applying the $\tau$-level perturbation kernel to a clean sample $\mathbf{x}^0 \sim p^{data}(\mathbf{x}^0)$. Importantly, as the kernel $p^{0\tau}$ is a known Gaussian distribution, this objective becomes a simple $L_2$ reconstruction loss,

$$\mathcal{L}(\theta) = \mathbb{E}\left[\|\mathbf{D}_\theta(\mathbf{x}^\tau, \tau) - \mathbf{x}^0\|^2\right], \tag{4}$$

with reparameterization $\mathbf{D}_\theta(\mathbf{x}^\tau, \tau) = \mathbf{S}_\theta(\mathbf{x}^\tau, \tau)\sigma^2(\tau) + \mathbf{x}^\tau$, where $\sigma(\tau)$ is the variance of the $\tau$-level perturbation kernel.

## 2.3 Diffusion for world modeling

The score-based diffusion model described in Section 2.2 provides an unconditional generative model of $p_{data}$. To serve as a world model, we need a conditional generative model of the environment dynamics, $p(\mathbf{x}_{t+1} \mid \mathbf{x}_{\leq t}, a_{\leq t})$, where we consider the general case of a POMDP, in which the Markovian state $s_t$ is unknown and can be approximated from past observations and actions. We can condition a diffusion model on this history, to estimate and generate the next observation directly, as shown in Figure 1. This modifies Equation 4 as follows,

$$\mathcal{L}(\theta) = \mathbb{E}\left[\|\mathbf{D}_\theta(\mathbf{x}_{t+1}^\tau, \tau, \mathbf{x}_{\leq t}^0, a_{\leq t}) - \mathbf{x}_{t+1}^0\|^2\right]. \tag{5}$$

During training, we sample a trajectory segment $\mathbf{x}_{\leq t}^0, a_{\leq t}, \mathbf{x}_{t+1}^0$ from the agent's replay dataset, and we obtain the noised next observation $\mathbf{x}_{t+1}^\tau \sim p^{0\tau}(\mathbf{x}_{t+1}^\tau \mid \mathbf{x}_{t+1}^0)$ by applying the $\tau$-level perturbation kernel. In summary, this diffusion process for world modeling resembles the standard diffusion process described in Section 2.2, with a score model conditioned on past observations and actions.

To sample the next observation, we iteratively solve the reverse SDE in Equation 2, as illustrated in Figure 1. While we can in principle resort to any ODE or SDE solver, there is an inherent trade-off between sampling quality and Number of Function Evaluations (NFE), that directly determines the inference cost of the diffusion world model (see Appendix A for more details).

# 3 Method

## 3.1 Practical choice of diffusion paradigm

Building on the background provided in Section 2, we now introduce DIAMOND as a practical realization of a diffusion-based world model. In particular, we now define the drift and diffusion coefficients $\mathbf{f}$ and $g$ introduced in Section 2.2, corresponding to a particular choice of diffusion paradigm. While DDPM (Ho et al., 2020) is an example of one such choice (as described in Appendix B) and would historically be the natural candidate, we instead build upon the EDM formulation proposed in Karras et al. (2022). The practical implications of this choice are discussed in Section 5.1. In what follows, we describe how we adapt EDM to build our diffusion-based world model.

We consider the perturbation kernel $p^{0\tau}(\mathbf{x}_{t+1}^\tau \mid \mathbf{x}_{t+1}^0) = \mathcal{N}(\mathbf{x}_{t+1}^\tau; \mathbf{x}_{t+1}^0, \sigma^2(\tau)\mathbf{I})$, where $\sigma(\tau)$ is a real-valued function of diffusion time called the noise schedule. This corresponds to setting the drift and diffusion coefficients to $\mathbf{f}(\mathbf{x}, \tau) = \mathbf{0}$ (affine) and $g(\tau) = \sqrt{2\dot{\sigma}(\tau)\sigma(\tau)}$.

We use the network preconditioning introduced by Karras et al. (2022) and so parameterize $\mathbf{D}_\theta$ in Equation 5 as the weighted sum of the noised observation and the prediction of a neural network $\mathbf{F}_\theta$,

$$\mathbf{D}_\theta(\mathbf{x}_{t+1}^\tau, y_t^\tau) = c_{\text{skip}}^\tau \mathbf{x}_{t+1}^\tau + c_{\text{out}}^\tau \mathbf{F}_\theta(c_{\text{in}}^\tau \mathbf{x}_{t+1}^\tau, y_t^\tau), \tag{6}$$

where for brevity we define $y_t^\tau := (c_{\text{noise}}^\tau, \mathbf{x}_{\leq t}^0, a_{\leq t})$ to include all conditioning variables.

The preconditioners $c_{\text{in}}^\tau$ and $c_{\text{out}}^\tau$ are selected to keep the network's input and output at unit variance for any noise level $\sigma(\tau)$, $c_{\text{noise}}^\tau$ is an empirical transformation of the noise level, and $c_{\text{skip}}^\tau$ is given in terms of $\sigma(\tau)$ and the standard deviation of the data distribution $\sigma_{data}$, as $c_{skip}^\tau = \sigma_{data}^2/(\sigma_{data}^2 + \sigma^2(\tau))$. These preconditioners are fully described in Appendix C.

Combining Equations 5 and 6 provides insight into the training objective of $\mathbf{F}_\theta$,

$$\mathcal{L}(\theta) = \mathbb{E}\left[\|\underbrace{\mathbf{F}_\theta(c_{\text{in}}^\tau \mathbf{x}_{t+1}^\tau, y_t^\tau)}_{\text{Network prediction}} - \underbrace{\frac{1}{c_{\text{out}}^\tau}(\mathbf{x}_{t+1}^0 - c_{\text{skip}}^\tau \mathbf{x}_{t+1}^\tau)}_{\text{Network training target}}\|^2\right]. \tag{7}$$

The network training target adaptively mixes signal and noise depending on the degradation level $\sigma(\tau)$. When $\sigma(\tau) \gg \sigma_{\text{data}}$, we have $c_{\text{skip}}^\tau \to 0$, and the training target for $\mathbf{F}_\theta$ is dominated by the clean signal $\mathbf{x}_{t+1}^0$. Conversely, when the noise level is low, $\sigma(\tau) \to 0$, we have $c_{\text{skip}}^\tau \to 1$, and the target becomes the difference between the clean and the perturbed signal, i.e. the added Gaussian noise. Intuitively, this prevents the training objective to become trivial in the low-noise regime. In practice, this objective is high variance at the extremes of the noise schedule, so Karras et al. (2022) sample the noise level $\sigma(\tau)$ from an empirically chosen log-normal distribution in order to concentrate the training around medium-noise regions, as described in Appendix C.

We use a standard U-Net 2D for the vector field $\mathbf{F}_\theta$ (Ronneberger et al., 2015), and we keep a buffer of $L$ past observations and actions that we use to condition the model. We concatenate these past observations to the next noisy observation channel-wise, and we input actions through adaptive group normalization layers (Zheng et al., 2020) in the residual blocks (He et al., 2015) of the U-Net.

As discussed in Section 2.3 and Appendix A, there are many possible sampling methods to generate the next observation from the trained diffusion model. While our codebase supports a variety of sampling schemes, we found Euler's method to be effective without incurring the cost of additional NFE required by higher order samplers, or the unnecessary complexity of stochastic sampling.

### 3.2 Reinforcement learning in imagination

Given the diffusion model from Section 3.1, we now complete our world model with a reward and termination model, required for training an RL agent in imagination. Since estimating the reward and termination are scalar prediction problems, we use a separate model $R_\psi$ consisting of standard CNN (LeCun et al., 1989; He et al., 2015) and LSTM (Hochreiter and Schmidhuber, 1997; Gers et al., 2000) layers to handle partial observability. The RL agent involves an actor-critic network parameterized by a shared CNN-LSTM with policy and value heads. The policy $\pi_\phi$ is trained with REINFORCE with a value baseline, and we use a Bellman error with $\lambda$-returns to train the value network $V_\phi$, similar to Micheli et al. (2023). We train the agent entirely in imagination as described in Section 2.1. The agent only interacts with the real environment for data collection. After each collection stage, the current world model is updated by training on all data collected so far. Then, the agent is trained with RL in the updated world model environment, and these steps are repeated. This procedure is detailed in Algorithm 1, and is similar to Kaiser et al. (2019); Hafner et al. (2020); Micheli et al. (2023). We provide architecture details, hyperparameters, and RL objectives in Appendices D, E, F, respectively.

## 4 Experiments

### 4.1 Atari 100k benchmark

For comprehensive evaluation of DIAMOND we use the established Atari 100k benchmark (Kaiser et al., 2019), consisting of 26 games that test a wide range of agent capabilities. For each game, an agent is only allowed to take 100k actions in the environment, which is roughly equivalent to 2 hours of human gameplay, to learn to play the game before evaluation. As a reference, unconstrained Atari agents are usually trained for 50 million steps, a 500 fold increase in experience. We trained DIAMOND from scratch for 5 random seeds on each game. Each run utilized around 12GB of VRAM and took approximately 2.9 days on a single Nvidia RTX 4090 (1.03 GPU years in total).

We compare with other recent methods training an agent entirely within a world model in Table 1, including STORM (Zhang et al., 2023), DreamerV3 (Hafner et al., 2023), IRIS (Micheli et al., 2023), TWM (Robine et al., 2023), and SimPle (Kaiser et al., 2019). A broader comparison to model-free and search-based methods, including BBF (Schwarzer et al., 2023) and EfficientZero (Ye et al., 2021), the current best performing methods on this benchmark, is provided in Appendix J. BBF and EfficientZero use techniques that are orthogonal and not directly comparable to our approach, such as using periodic network resets in combination with hyperparameter scheduling for BBF, and computationally expensive lookahead Monte-Carlo tree search for EfficientZero. Combining these additional components with our world model would be an interesting direction for future work.

### 4.2 Results on the Atari 100k benchmark

Table 1 provides scores for all games, and the mean and interquartile mean (IQM) of human-normalized scores (HNS) (Wang et al., 2016). Following the recommendations of Agarwal et al. (2021) on the limitations of point estimates, we provide stratified bootstrap confidence intervals for the mean and IQM in Figure 2, as well as performance profiles and additional metrics in Appendix H.

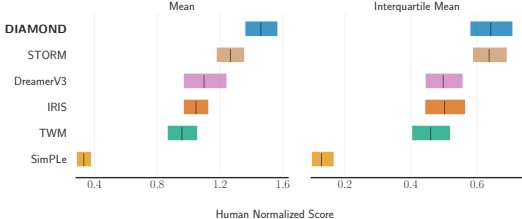

Figure 2: Mean and interquartile mean human normalized scores. DIAMOND, in blue, obtains a mean HNS of 1.46 and an IQM of 0.64.

Table 1: Returns on the 26 games of the Atari 100k benchmark after 2 hours of real-time experience, and human-normalized aggregate metrics. Bold numbers indicate the best performing methods. DIAMOND notably outperforms other world model baselines in terms of mean score over 5 seeds.

| Game | Random | Human | SimPLe | TWM | IRIS | DreamerV3 | STORM | DIAMOND (ours) |
|---|---|---|---|---|---|---|---|---|
| Alien | 227.8 | 7127.7 | 616.9 | 674.6 | 420.0 | 959.0 | **983.6** | 744.1 |
| Amidar | 5.8 | 1719.5 | 74.3 | 121.8 | 143.0 | 139.0 | 204.8 | **225.8** |
| Assault | 222.4 | 742.0 | 527.2 | 682.6 | 1524.4 | 706.0 | 801.0 | **1526.4** |
| Asterix | 210.0 | 8503.3 | 1128.3 | 1116.6 | 853.6 | 932.0 | 1028.0 | **3698.5** |
| BankHeist | 14.2 | 753.1 | 34.2 | 466.7 | 53.1 | **649.0** | 641.2 | 19.7 |
| BattleZone | 2360.0 | 37187.5 | 4031.2 | 5068.0 | 13074.0 | 12250.0 | **13540.0** | 4702.0 |
| Boxing | 0.1 | 12.1 | 7.8 | 77.5 | 70.1 | 78.0 | 79.7 | **86.9** |
| Breakout | 1.7 | 30.5 | 16.4 | 20.0 | 83.7 | 31.0 | 15.9 | **132.5** |
| ChopperCommand | 811.0 | 7387.8 | 979.4 | 1697.4 | 1565.0 | 420.0 | **1888.0** | 1369.8 |
| CrazyClimber | 10780.5 | 35829.4 | 62583.6 | 71820.4 | 59324.2 | 97190.0 | 66776.0 | **99167.8** |
| DemonAttack | 152.1 | 1971.0 | 208.1 | 350.2 | **2034.4** | 303.0 | 164.6 | 288.1 |
| Freeway | 0.0 | 29.6 | 16.7 | 24.3 | 31.1 | 0.0 | **33.5** | 33.3 |
| Frostbite | 65.2 | 4334.7 | 236.9 | **1475.6** | 259.1 | 909.0 | 1316.0 | 274.1 |
| Gopher | 257.6 | 2412.5 | 596.8 | 1674.8 | 2236.1 | 3730.0 | **8239.6** | 5897.9 |
| Hero | 1027.0 | 30826.4 | 2656.6 | 7254.0 | 7037.4 | **11161.0** | 11044.3 | 5621.8 |
| Jamesbond | 29.0 | 302.8 | 100.5 | 362.4 | 462.7 | 445.0 | **509.0** | 427.4 |
| Kangaroo | 52.0 | 3035.0 | 51.2 | 1240.0 | 838.2 | 4098.0 | 4208.0 | **5382.2** |
| Krull | 1598.0 | 2665.5 | 2204.8 | 6349.2 | 6616.4 | 7782.0 | 8412.6 | **8610.1** |
| KungFuMaster | 258.5 | 22736.3 | 14862.5 | 24554.6 | 21759.8 | 21420.0 | **26182.0** | 18713.6 |
| MsPacman | 307.3 | 6951.6 | 1480.0 | 1588.4 | 999.1 | 1327.0 | **2673.5** | 1958.2 |
| Pong | -20.7 | 14.6 | 12.8 | 18.8 | 14.6 | 18.0 | 11.3 | **20.4** |
| PrivateEye | 24.9 | 69571.3 | 35.0 | 86.6 | 100.0 | 882.0 | **7781.0** | 114.3 |
| Qbert | 163.9 | 13455.0 | 1288.8 | 3330.8 | 745.7 | 3405.0 | **4522.5** | 4499.3 |
| RoadRunner | 11.5 | 7845.0 | 5640.6 | 9109.0 | 9614.6 | 15565.0 | 17564.0 | **20673.2** |
| Seaquest | 68.4 | 42054.7 | 683.3 | **774.4** | 661.3 | 618.0 | 525.2 | 551.2 |
| UpNDown | 533.4 | 11693.2 | 3350.3 | **15981.7** | 3546.2 | 9234.0 | 7985.0 | 3856.3 |
| #Superhuman ($\uparrow$) | 0 | N/A | 1 | 8 | 10 | 9 | 10 | **11** |
| Mean ($\uparrow$) | 0.000 | 1.000 | 0.332 | 0.956 | 1.046 | 1.097 | 1.266 | **1.459** |
| IQM ($\uparrow$) | 0.000 | 1.000 | 0.130 | 0.459 | 0.501 | 0.497 | **0.636** | **0.641** |

Our results demonstrate that DIAMOND performs strongly across the benchmark, outperforming human players on 11 games, and achieving a superhuman mean HNS of 1.46, a new best among agents trained entirely within a world model. DIAMOND also achieves an IQM on par with STORM, and greater than all other baselines. We find that DIAMOND performs particularly well on environments where capturing small details is important, such as *Asterix*, *Breakout* and *Road Runner*. We provide further qualitative analysis of the visual quality of the world model in Section 5.3.

## 5 Analysis

### 5.1 Choice of diffusion framework

As explained in Section 2, we could in principle use any diffusion model variant in our world model. While DIAMOND utilizes EDM (Karras et al., 2022) as described in Section 3, DDPM (Ho et al., 2020) would also be a natural candidate, having been used in many image generation applications (Rombach et al., 2022; Nichol and Dhariwal, 2021). We justify this design decision in this section.

To provide a fair comparison of DDPM with our EDM implementation, we train both variants with the same network architecture, on a shared static dataset of 100k frames collected with an expert policy on the game *Breakout*. As discussed in Section 2.3, the number of denoising steps is directly related to the inference cost of the world model, and so fewer steps will reduce the cost of training an agent on imagined trajectories. Ho et al. (2020) use a thousand denoising steps, and Rombach et al. (2022) employ hundreds steps for Stable Diffusion. However, for our world model to be computationally comparable with other world model baselines (such as IRIS which requires 16 NFE for each timestep), we need at most tens of denoising steps, and preferably fewer. Unfortunately, if the number of denoising steps is set too low, the visual quality will degrade, leading to compounding error.

To investigate the stability of the diffusion variants, we display imagined trajectories generated autoregressively up to $t = 1000$ timesteps in Figure 3, for different numbers of denoising steps $n \leq 10$. We see that using DDPM (Figure 3a) in this regime leads to severe compounding error, causing the world model to quickly drift out of distribution. In contrast, the EDM-based diffusion world model (Figure 3b) appears much more stable over long time horizons, even for a single denoising step. A quantitative analysis of this compounding error is provided in Appendix K.

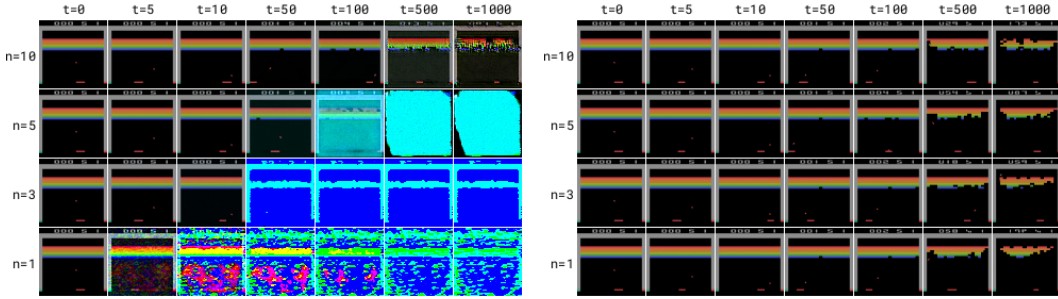

(a) DDPM-based world model trajectories.   (b) EDM-based world model trajectories.

Figure 3: Imagined trajectories with diffusion world models based on DDPM (left) and EDM (right). The initial observation at $t = 0$ is common, and each row corresponds to a decreasing number of denoising steps $n$. We observe that DDPM-based generation suffers from compounding error, and that the smaller the number of denoising steps, the faster the error accumulates. In contrast, our EDM-based world model appears much more stable, even for $n = 1$.

This surprising result is a consequence of the improved training objective described in Equation 7, compared to the simpler noise prediction objective employed by DDPM. While predicting the noise works well for intermediate noise levels, this objective causes the model to learn the identity function when the noise is dominant ($\sigma_{noise} \gg \sigma_{data} \implies \xi_\theta(\mathbf{x}_{t+1}^\tau, y_t^\tau) \to \mathbf{x}_{t+1}^\tau$), where $\xi_\theta$ is the noise prediction network of DDPM. This gives a poor estimate of the score function at the beginning of the sampling procedure, which degrades the generation quality and leads to compounding error.

In contrast, the adaptive mixing of signal and noise employed by EDM, described in Section 3.1, means that the model is trained to predict the clean image when the noise is dominant ($\sigma_{noise} \gg \sigma_{data} \implies \mathbf{F}_\theta(\mathbf{x}_{t+1}^\tau, y_t^\tau) \to \mathbf{x}_{t+1}^0$). This gives a better estimate of the score function in the absence of signal, so the model is able to produce higher quality generations with fewer denoising steps, as illustrated in Figure 3b.

## 5.2 Choice of the number of denoising steps

While we found that our EDM-based world model was very stable with just a single denoising step, as shown for *Breakout* in the last row of Figure 3b, we discuss here how this choice would limit the visual quality of the model in some cases. We provide more a quantitative analysis in Appendix L.

As discussed in Section 2.2, our score model is equivalent to a denoising autoencoder (Vincent et al., 2008) trained with an $L_2$ reconstruction loss. The optimal single-step prediction is thus the expectation over possible reconstructions for a given noisy input, which can be out of distribution if this posterior distribution is multimodal. While some games like *Breakout* have deterministic transitions that can be accurately modeled with a single denoising step (see Figure 3b), in some other games partial observability gives rise to multimodal observation distributions. In this case, an iterative solver is necessary to drive the sampling procedure towards a particular mode, as illustrated in the game *Boxing* in Figure 4. As a result, we therefore set $n = 3$ in all of our experiments.

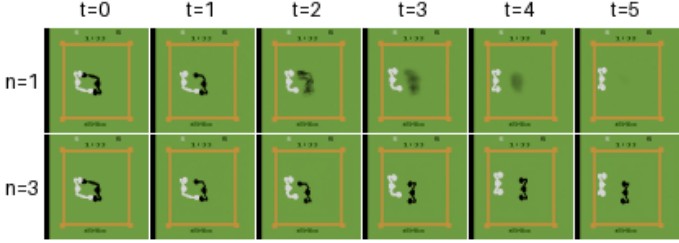

Figure 4: Single-step (top row) versus multi-step (bottom row) sampling in *Boxing*. Movements of the black player are unpredictable, so that single-step denoising interpolates between possible outcomes and results in blurry predictions. In contrast, multi-step sampling produces a crisp image by driving the generation towards a particular mode. Interestingly, the policy controls the white player, so his actions are known to the world model. This information removes any ambiguity, and so we observe that both single-step and multi-step sampling correctly predict the white player's position.

### 5.3 Qualitative visual comparison with IRIS

We now compare to IRIS (Micheli et al., 2023), a well-established world model that uses a discrete autoencoder (Van Den Oord et al., 2017) to convert images to discrete tokens, and composes these tokens over time with an autoregressive transformer (Radford et al., 2019). For fair comparison, we train both world models on the same static datasets of 100k frames collected with expert policies. This comparison is displayed in Figure 5 below.

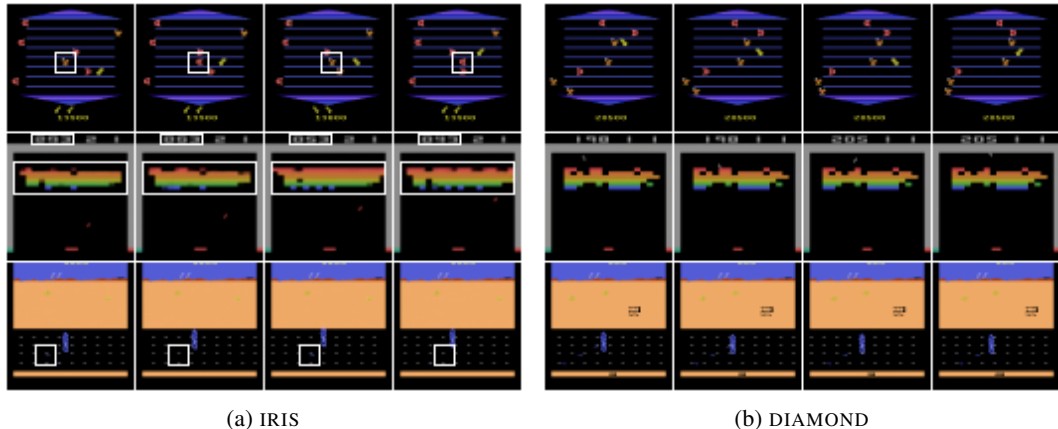

(a) IRIS                                                                                    (b) DIAMOND

Figure 5: Consecutive frames imagined with IRIS (left) and DIAMOND (right). The white boxes highlight inconsistencies between frames, which we see only arise in trajectories generated with IRIS. In *Asterix* (top row), an enemy (orange) becomes a reward (red) in the second frame, before reverting to an enemy in the third, and again to a reward in the fourth. In *Breakout* (middle row), the bricks and score are inconsistent between frames. In *Road Runner* (bottom row), the rewards (small blue dots on the road) are inconsistently rendered between frames. None of these inconsistencies occur with DIAMOND. In *Breakout*, the score is even reliably updated by +7 when a red brick is broken[3].

We see in Figure 5 that the trajectories imagined by DIAMOND are generally of higher visual quality and more faithful to the true environment compared to the trajectories imagined by IRIS. In particular, the trajectories generated by IRIS contain visual inconsistencies between frames (highlighted by white boxes), such as enemies being displayed as rewards and vice-versa. These inconsistencies may only represent a few pixels in the generated images, but can have significant consequences for reinforcement learning. For example, since an agent should generally target rewards and avoid enemies, these small visual discrepancies can make it more challenging to learn an optimal policy.

These improvements in the consistency of visual details are generally reflected by greater agent performance on these games, as shown in Table 1. Since the agent component of these methods is similar, this improvement can likely be attributed to the world model.

Finally, we note that this improvement is not simply the result of increased computation. Both world models are rendering frames at the same resolution ($64 \times 64$), and DIAMOND requires only 3 NFE per frame compared to 16 NFE per frame for IRIS. This is further reflected by the fact that DIAMOND has significantly fewer parameters and takes less time to train than IRIS, as provided in Appendix H.

## 6  Scaling the diffusion world model to *Counter-Strike: Global Offensive*[4]

To investigate the ability of DIAMOND's diffusion world model to learn to model more complex 3D environments, we train the world model in isolation on static data from the popular video game *Counter-Strike: Global Offensive* (CS:GO). We use the *Online* dataset of 5.5M frames (95 hours) of online human gameplay captured at 16Hz from the map *Dust II* by Pearce and Zhu (2022). We randomly hold out 0.5M frames (corresponding to 500 episodes, or 8 hours) for testing, and use the remaining 5M frames (87 hours) for training. There is no reinforcement learning agent or online data collection involved in these experiments.

---

[3]`https://en.wikipedia.org/wiki/Breakout_(video_game)#Gameplay`
[4]This section was added after NeurIPS acceptance, following community interest in later CS:GO experiments.

To reduce the computational cost, we reduce the resolution from $(280 \times 150)$ to $(56 \times 30)$ for world modeling. We then introduce a second, smaller diffusion model as an upsampler to improve the generated images at the original resolution (Saharia et al., 2022b). We scale the channels of the U-Net to increase the number of parameters from 4M for our Atari models to 381M for our CS:GO model (including 51M for the upsampler). The combined model was trained for 12 days on an RTX 4090.

Finally, we introduce stochastic sampling and increase the number of denoising steps for the upsampler to 10, which we found to improve the resulting visual quality of the generations, while keeping the dynamics model the same (in particular, still using only 3 denoising steps). This enables a reasonable tradeoff between visual quality and inference cost, with the model running at 10Hz on an RTX 3090. Typical generations of the model are provided in Figure 6 below.

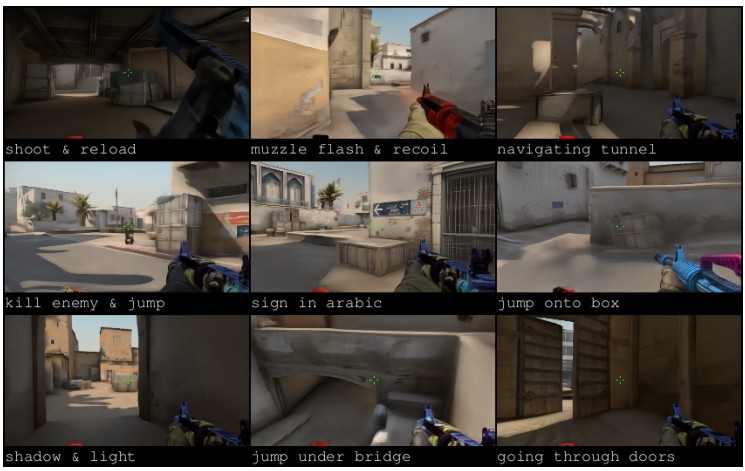

Figure 6: Images captured from people playing with keyboard and mouse inside DIAMOND's diffusion world model. This model was trained on 87 hours of static *Counter-Strike: Global Offensive* (CS:GO) gameplay (Pearce and Zhu, 2022) to produce an interactive neural game engine for the popular in-game map, *Dust II*. Best viewed as videos at `https://diamond-wm.github.io`.

We find the model is able to generate stable trajectories over hundreds of timesteps, although is more likely to drift out-of-distribution in less frequently visited areas of the map. Due to the limited memory of the model, approaching walls or losing visibility may cause the model to forget the current state and instead generate a new weapon or area of map. Interestingly, we find the model wrongly enables successive jumps by generalizing the effect of a jump on the geometry of the scene, since multiple jumps do not appear often enough in the training gameplay for the model to learn that mid-air jumps should be ignored. We expect scaling the model and data to address many of these limitations, with the exception of the memory of the model. Quantitative measurements of the capabilities of the CS:GO world model and attempts to address these limitations are left to future work.

## 7  Related work

**World models.** The idea of reinforcement learning (RL) in the imagination of a neural network world model was introduced by Ha and Schmidhuber (2018). SimPLe (Kaiser et al., 2019) applied world models to Atari, and introduced the Atari 100k benchmark to focus on sample efficiency. Dreamer (Hafner et al., 2020) introduced RL from the latent space of a recurrent state space model (RSSM). DreamerV2 (Hafner et al., 2021) demonstrated that using discrete latents could help to reduce compounding error, and DreamerV3 (Hafner et al., 2023) was able to achieve human-level performance on a wide range of domains with fixed hyperparameters. TWM (Robine et al., 2023) adapts DreamerV2's RSSM to use a transformer architecture, while STORM (Zhang et al., 2023) adapts DreamerV3 in a similar way but with a different tokenization approach. Alternatively, IRIS (Micheli et al., 2023) builds a language of image tokens with a discrete autoencoder, and composes these tokens over time with an autoregressive transformer.

**Generative vision models.** There are parallels between these world models and image generation models which suggests that developments in generative vision models could provide benefits to world modeling. Following the rise of transformers in natural language processing (Vaswani et al.,

2017; Devlin et al., 2018; Radford et al., 2019), VQGAN (Esser et al., 2021) and DALL·E (Ramesh et al., 2021) convert images to discrete tokens with discrete autoencoders (Van Den Oord et al., 2017), and leverage the sequence modeling abilities of autoregressive transformers to build powerful text-to-image generative models. Concurrently, diffusion models (Sohl-Dickstein et al., 2015; Ho et al., 2020; Song et al., 2020) gained traction (Dhariwal and Nichol, 2021; Rombach et al., 2022), and have become a dominant paradigm for high-resolution image generation (Saharia et al., 2022a; Ramesh et al., 2022; Podell et al., 2023).

The same trends have taken place in the recent developments of video generation methods. VideoGPT (Yan et al., 2021) provides a minimal video generation architecture by combining a discrete autoencoder with an autoregressive transformer. Godiva (Wu et al., 2021) enables text conditioning with promising generalization. Phenaki (Villegas et al., 2023) allows arbitrary length video generation with sequential prompt conditioning. TECO (Yan et al., 2023) improves upon autoregressive modeling by using MaskGit (Chang et al., 2022), and enables longer temporal dependencies by compressing input sequence embeddings. Diffusion models have also seen a resurgence in video generation using 3D U-Nets to provide high quality but short-duration video (Singer et al., 2023; Bar-Tal et al., 2024). Recently, transformer-based diffusion models such as DiT (Peebles and Xie, 2023) and Sora (Brooks et al., 2024) have shown improved scalability for both image and video generation, respectively.

**Diffusion for reinforcement learning.** There has also been much interest in combining diffusion models with reinforcement learning. This includes taking advantage of the flexibility of diffusion models as a policy (Wang et al., 2022; Ajay et al., 2022; Pearce et al., 2023), as planners (Janner et al., 2022; Liang et al., 2023), as reward models (Nuti et al., 2023), and trajectory modeling for data augmentation in offline RL (Lu et al., 2023; Ding et al., 2024; Jackson et al., 2024). DIAMOND represents the first use of diffusion models as world models for learning online in imagination.

**Generative game engines.** Playable games running entirely on neural networks have recently been growing in scope. *GameGAN* (Kim et al., 2020) learns generative models of games using a GAN (Goodfellow et al., 2014) while Bamford and Lucas (2020) use a Neural GPU (Kaiser and Sutskever, 2015). Concurrent work includes *Genie* (Bruce et al., 2024), which generates playable platformer environments from image prompts, and *GameNGen* (Valevski et al., 2024), which similarly leverages a diffusion model to obtain a high resolution simulator of the game DOOM, but at a larger scale.

## 8  Limitations

We identify three main limitations of our work for future research. First, our main evaluation is focused on discrete control environments, and applying DIAMOND to the continuous domain may provide additional insights. Second, the use of frame stacking for conditioning is a minimal mechanism to provide a memory of past observations. Integrating an autoregressive transformer over environment time, using an approach such as Peebles and Xie (2023), would enable longer-term memory and better scalability. We include an initial investigation into a potential cross-attention architecture in Appendix M, but found frame-stacking more effective in our early experiments. Third, we leave potential integration of the reward/termination prediction into the diffusion model for future work, since combining these objectives and extracting representations from a diffusion model is not trivial (Luo et al., 2023; Xu et al., 2023) and would make our world model unnecessarily complex.

## 9  Conclusion and Broader Impact

We have introduced DIAMOND, a reinforcement learning agent trained in a diffusion world model. We explained the key design choices we made to adapt diffusion for world modeling and to make our world model stable over long time horizons with a low number of denoising steps. DIAMOND achieves a mean human normalized score of $1.46$ on the well-established Atari 100k benchmark; a new best among agents trained entirely within a world model. We analyzed our improved performance in some games and found that it likely follows from better modeling of critical visual details. We further demonstrated DIAMOND's diffusion world model can successfully model 3D environments and serve as a real-time neural game engine by training on static *Counter-Strike: Global Offensive* gameplay.

World models constitute a promising direction to address sample efficiency and safety concerns associated with training agents in the real world. However, imperfections in the world model may lead to suboptimal or unexpected agent behaviors. We hope that the development of more faithful and interactive world models will contribute to broader efforts to further reduce these risks.

## Acknowledgments and Disclosure of Funding

We would like to thank Andrew Foong, Bálint Máté, Clément Vignac, Maxim Peter, Pedro Sanchez, Rich Turner, Stéphane Nguyen, Tom Lee, Trevor McInroe and Weipu Zhang for insightful discussions and comments. Adam and Eloi met during an internship at Microsoft Research Cambridge, and would like to thank the Game Intelligence team, including Anssi Kanervisto, Dave Bignell, Gunshi Gupta, Katja Hofmann, Lukas Schäfer, Raluca Georgescu, Sam Devlin, Sergio Valcarcel Macua, Shanzheng Tan, Tabish Rashid, Tarun Gupta, Tim Pearce, and Yuhan Cao, for their support in the early stages of this project, and a great summer.

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

## A   Sampling observations in DIAMOND

We describe here how we sample an observation $\mathbf{x}_t^0$ from our diffusion world model. We initialize the procedure with a noisy observation $\mathbf{x}_t^{\mathcal{T}} \sim p^{prior}$, and iteratively solve the reverse SDE in Equation 2 from $\tau = \mathcal{T}$ to $\tau = 0$, using the learned score model $\mathbf{S}_\theta(\mathbf{x}_t^\tau, \tau, \mathbf{x}_{<t}^0, a_{<t})$ conditioned on past observations $\mathbf{x}_{<t}^0$ and actions $a_{<t}$. This procedure is illustrated in Figure 1.

In fact, there are many possible sampling methods for a given learned score model $\mathbf{S}_\theta$ (Karras et al., 2022). Notably, Song et al. (2020) introduce a corresponding "probability flow" ordinary differential equation (ODE), with marginals equivalent to the stochastic process described in Section 2.2. In that case, the solving procedure is deterministic, and the only randomness comes from sampling the initial condition. In practice, this means that for a given score model, we can resort to any ODE or SDE solver, from simple first order methods like Euler (deterministic) and Euler–Maruyama (stochastic) schemes, to higher-order methods like Heun's method (Ascher and Petzold, 1998).

Regardless of the choice of solver, each step introduces truncation errors, resulting from the local score approximation and the discretization of the continuous process. Higher order samplers may reduce this truncation error, but come at the cost of additional Number of Function Evaluations (NFE) – how many forward passes of the network are required to generate a sample. This local error generally scales superlinearly with respect to the step size (for instance Euler's method is $\mathcal{O}(h^2)$ for step size $h$), so increasing the number of denoising steps improves the visual quality of the generated next frame. Therefore, there is a trade-off between visual quality and NFE that directly determines the inference cost of the diffusion world model.

## B   Link between DDPM and continuous-time score-based diffusion models

Denoising Diffusion Probabilistic Models (DDPM, Ho et al. (2020)) can be described as a discrete version of the diffusion process introduced in Section 2.2, as described in Song et al. (2020). The discrete forward process is a Markov chain characterized by a discrete noise schedule $0 < \beta_1, \ldots, \beta_i, \ldots \beta_N < 1$, and a variance-preserving Gaussian transition kernel,

$$p(\mathbf{x}^i|\mathbf{x}^{i-1}) = \mathcal{N}(\mathbf{x}^i; \sqrt{1 - \beta_i}\mathbf{x}^{i-1}, \beta_i\mathbf{I}). \tag{8}$$

In the continuous time limit $N \to \infty$, the Markov chain becomes a diffusion process, and the discrete noise schedule becomes a time-dependent function $\beta(\tau)$. This diffusion process can be described by an SDE with drift coefficient $\mathbf{f}(\mathbf{x}, \tau) = -\frac{1}{2}\beta(\tau)\mathbf{x}$ and diffusion coefficient $g(\tau) = \sqrt{\beta(\tau)}$ (Song et al., 2020).

## C   EDM network preconditioners and training

Karras et al. (2022) use the following preconditioners for normalization and rescaling purposes (as mentioned in Section 3.1) to improve network training:

$$c_{in}^\tau = \frac{1}{\sqrt{\sigma(\tau)^2 + \sigma_{data}^2}} \tag{9}$$

$$c_{out}^\tau = \frac{\sigma(\tau)\sigma_{data}}{\sqrt{\sigma(\tau)^2 + \sigma_{data}^2}} \tag{10}$$

$$c_{noise}^\tau = \frac{1}{4}\log(\sigma(\tau)) \tag{11}$$

$$c_{skip}^\tau = \frac{\sigma_{data}^2}{\sigma_{data}^2 + \sigma^2(\tau)}, \tag{12}$$

where $\sigma_{data} = 0.5$.

The noise parameter $\sigma(\tau)$ is sampled to maximize the effectiveness of training as follows:

$$\log(\sigma(\tau)) \sim \mathcal{N}(P_{mean}, P_{std}^2), \tag{13}$$

where $P_{mean} = -0.4, P_{std} = 1.2$. Refer to Karras et al. (2022) for an in-depth analysis.

# D  Model architectures

The diffusion model $\mathbf{D}_\theta$ is a standard U-Net 2D (Ronneberger et al., 2015), conditioned on the last 4 frames and actions, as well as the diffusion time $\tau$. We use frame stacking for observation conditioning, and adaptive group normalization (Zheng et al., 2020) for action and diffusion time conditioning.

The reward/termination model $R_\psi$ layers are shared except for the final prediction heads. The model takes as input a sequence of frames and actions, and forwards it through convolutional residual blocks (He et al., 2015) followed by an LSTM cell (Mnih et al., 2016; Hochreiter and Schmidhuber, 1997; Gers et al., 2000). Before starting the imagination procedure, we burn-in (Kapturowski et al., 2018) the conditioning frames and actions to initialize the hidden and cell states of the LSTM.

The weights of the policy $\pi_\phi$ and value network $V_\phi$ are shared except for the last layer. In the following, we refer to $(\pi, V)_\phi$ as the "actor-critic" network, even though $V$ is technically a state-value network, not a critic. This network takes as input a frame, and forwards it through convolutional trunk followed by an LSTM cell. The convolutional trunk consists of four residual blocks and 2x2 max-pooling with stride 2. The main path of the residual blocks consists of a group normalization (Wu and He, 2018) layer, a SiLU activation (Elfwing et al., 2018), and a 3x3 convolution with stride 1 and padding 1. Before starting the imagination procedure, we burn-in the conditioning frames to initialize the hidden and cell states of the LSTM.

Please refer to Table 2 below for hyperparameter values, and to Algorithm 1 for a detailed summary of the training procedure.

Table 2: Architecture details for DIAMOND.

| Hyperparameter | Value |
|---|:---:|
| **Diffusion Model ($\mathbf{D}_\theta$)** | |
| Observation conditioning mechanism | Frame stacking |
| Action conditioning mechanism | Adaptive Group Normalization |
| Diffusion time conditioning mechanism | Adaptive Group Normalization |
| Residual blocks layers | [2, 2, 2, 2] |
| Residual blocks channels | [64, 64, 64, 64] |
| Residual blocks conditioning dimension | 256 |
| | |
| **Reward/Termination Model ($R_\psi$)** | |
| Action conditioning mechanisms | Adaptive Group Normalization |
| Residual blocks layers | [2, 2, 2, 2] |
| Residual blocks channels | [32, 32, 32, 32] |
| Residual blocks conditioning dimension | 128 |
| LSTM dimension | 512 |
| | |
| **Actor-Critic Model ($\pi_\phi$ and $V_\phi$)** | |
| Residual blocks layers | [1, 1, 1, 1] |
| Residual blocks channels | [32, 32, 64, 64] |
| LSTM dimension | 512 |

# E   Training hyperparameters

Table 3: Hyperparameters for DIAMOND.

| Hyperparameter | Value |
|---|---|
| **Training loop** | |
| Number of epochs | 1000 |
| Training steps per epoch | 400 |
| Batch size | 32 |
| Environment steps per epoch | 100 |
| Epsilon (greedy) for collection | 0.01 |
| | |
| **RL hyperparameters** | |
| Imagination horizon ($H$) | 15 |
| Discount factor ($\gamma$) | 0.985 |
| Entropy weight ($\eta$) | 0.001 |
| $\lambda$-returns coefficient ($\lambda$) | 0.95 |
| | |
| **Sequence construction during training** | |
| For $\mathbf{D}_\theta$, number of conditioning observations and actions ($L$) | 4 |
| For $R_\psi$, burn-in length ($B_R$), set to $L$ in practice | 4 |
| For $R_\psi$, training sequence length ($B_R + H$) | 19 |
| For $\pi_\phi$ and $V_\phi$, burn-in length ($B_{\pi,V}$), set to $L$ in practice | 4 |
| | |
| **Optimization** | |
| Optimizer | AdamW |
| Learning rate | 1e-4 |
| Epsilon | 1e-8 |
| Weight decay ($\mathbf{D}_\theta$) | 1e-2 |
| Weight decay ($R_\psi$) | 1e-2 |
| Weight decay ($\pi_\phi$ and $V_\phi$) | 0 |
| | |
| **Diffusion Sampling** | |
| Method | Euler |
| Number of steps | 3 |
| | |
| **Environment** | |
| Image observation dimensions | 64×64×3 |
| Action space | Discrete (up to 18 actions) |
| Frameskip | 4 |
| Max noop | 30 |
| Termination on life loss | True |
| Reward clipping | $\{-1, 0, 1\}$ |

# F Reinforcement learning objectives

In what follows, we note $\mathbf{x}_t$, $r_t$ and $d_t$ the observations, rewards, and boolean episode terminations predicted by our world model. We note $H$ the imagination horizon, $V_\phi$ the value network, $\pi_\phi$ the policy network, and $a_t$ the actions taken by the policy within the world model.

We use $\lambda$-returns to balance bias and variance as the regression target for the value network. Given an imagined trajectory of length $H$, we can define the $\lambda$-return recursively as follows,

$$\Lambda_t = \begin{cases} r_t + \gamma(1 - d_t)\Big[(1 - \lambda)V_\phi(\mathbf{x}_{t+1}) + \lambda\Lambda_{t+1}\Big] & \text{if} \quad t < H \\ V_\phi(\mathbf{x}_H) & \text{if} \quad t = H. \end{cases} \tag{14}$$

The value network $V_\phi$ is trained to minimize $\mathcal{L}_V(\phi)$, the expected squared difference with $\lambda$-returns over imagined trajectories,

$$\mathcal{L}_V(\phi) = \mathbb{E}_{\pi_\phi}\left[\sum_{t=0}^{H-1} \left(V_\phi(\mathbf{x}_t) - \text{sg}(\Lambda_t)\right)^2\right], \tag{15}$$

where $\text{sg}(\cdot)$ denotes the gradient stopping operation, meaning that the target is a constant in the gradient-based optimization, as classically established in the literature (Mnih et al., 2015; Hafner et al., 2021; Micheli et al., 2023).

As we can generate large amounts of on-policy trajectories in imagination, we use a simple RE-INFORCE objective to train the policy, with the value $V_\phi(\mathbf{x}_t)$ as a baseline to reduce the variance of the gradients (Sutton and Barto, 2018). The policy is trained to minimize the following objective, combining REINFORCE and a weighted entropy maximization objective to maintain sufficient exploration,

$$\mathcal{L}_\pi(\phi) = -\mathbb{E}_{\pi_\phi}\left[\sum_{t=0}^{H-1} \log\left(\pi_\phi\left(a_t \mid \mathbf{x}_{\leq t}\right)\right) \text{sg}\left(\Lambda_t - V_\phi\left(\mathbf{x}_t\right)\right) + \eta\, \mathcal{H}\left(\pi_\phi\left(a_t \mid \mathbf{x}_{\leq t}\right)\right)\right]. \tag{16}$$

# G   DIAMOND algorithm

We summarize the overall training procedure of DIAMOND in Algorithm 1 below. We denote as $\mathcal{D}$ the replay dataset where the agent stores data collected from the real environment, and other notations are introduced in previous sections or are self-explanatory.

---

**Algorithm 1:** DIAMOND

---

**Procedure** `training_loop()`:
  **for** *epochs* **do**
      `collect_experience(`*steps_collect*`)`
      **for** *steps_diffusion_model* **do**
        `update_diffusion_model()`
      **for** *steps_reward_end_model* **do**
        `update_reward_end_model()`
      **for** *steps_actor_critic* **do**
        `update_actor_critic()`

**Procedure** `collect_experience(`$n$`)`:
  $\mathbf{x}_0^0 \leftarrow$ `env.reset()`
  **for** $t = 0$ **to** $n - 1$ **do**
      Sample $a_t \sim \pi_\phi(a_t \mid \mathbf{x}_t^0)$
      $\mathbf{x}_{t+1}^0, r_t, d_t \leftarrow$ `env.step(`$a_t$`)`
      $\mathcal{D} \leftarrow \mathcal{D} \cup \{\mathbf{x}_t^0, a_t, r_t, d_t\}$
      **if** $d_t = 1$ **then**
        $\mathbf{x}_{t+1}^0 \leftarrow$ `env.reset()`

**Procedure** `update_diffusion_model()`:
  Sample sequence $(\mathbf{x}_{t-L+1}^0, a_{t-L+1}, \ldots, \mathbf{x}_t^0, a_t, \mathbf{x}_{t+1}^0) \sim \mathcal{D}$
  Sample $\log(\sigma) \sim \mathcal{N}(P_{mean}, P_{std}^2)$ `// log-normal sigma distribution from EDM`
  Define $\tau := \sigma$          `// default identity schedule from EDM`
  Sample $\mathbf{x}_{t+1}^\tau \sim \mathcal{N}(\mathbf{x}_{t+1}^0, \sigma^2 \mathbf{I})$      `// Add independent Gaussian noise`
  Compute $\hat{\mathbf{x}}_{t+1}^0 = \mathbf{D}_\theta(\mathbf{x}_{t+1}^\tau, \tau, \mathbf{x}_{t-L+1}^0, a_{t-L+1}, \ldots, \mathbf{x}_t^0, a_t)$
  Compute reconstruction loss $\mathcal{L}(\theta) = \|\hat{\mathbf{x}}_{t+1}^0 - \mathbf{x}_{t+1}^0\|^2$
  Update $\mathbf{D}_\theta$

**Procedure** `update_reward_end_model()`:
  Sample indexes $\mathcal{I} := \{t, \ldots, t + L + H - 1\}$ `// burn-in + imagination horizon`
  Sample sequence $(\mathbf{x}_i^0, a_i, r_i, d_i)_{i \in \mathcal{I}} \sim \mathcal{D}$
  Initialize $h = c = 0$          `// LSTM hidden and cell states`
  **for** $i \in \mathcal{I}$ **do**
      Compute $\hat{r}_i, \hat{d}_i, h, c = R_\psi(\mathbf{x}_i, a_i, h, c)$
  Compute $\mathcal{L}(\psi) = \sum_{i \in \mathcal{I}} \mathrm{CE}(\hat{r}_i, \mathrm{sign}(r_i)) + \mathrm{CE}(\hat{d}_i, d_i)$ `// CE: cross-entropy loss`
  Update $R_\psi$

**Procedure** `update_actor_critic()`:
  Sample initial buffer $(\mathbf{x}_{t-L+1}^0, a_{t-L+1}, \ldots, \mathbf{x}_t^0) \sim \mathcal{D}$
  Burn-in buffer with $R_\psi$, $\pi_\phi$ and $V_\phi$ to initialize LSTM states
  **for** $i = t$ **to** $t + H - 1$ **do**
      Sample $a_i \sim \pi_\phi(a_i \mid \mathbf{x}_i^0)$
      Sample reward $r_i$ and termination $d_i$ with $R_\psi$
      Sample next observation $\mathbf{x}_{i+1}^0$ by simulating reverse diffusion process with $\mathbf{D}_\theta$
  Compute $V_\phi(\mathbf{x}_i)$ for $i = t, \ldots, t + H$
  Compute RL losses $\mathcal{L}_V(\phi)$ and $\mathcal{L}_\pi(\phi)$
  Update $\pi_\phi$ and $V_\phi$

---

## H Additional performance comparisons

We provide performance profiles (Agarwal et al., 2021) for DIAMOND and baselines below.

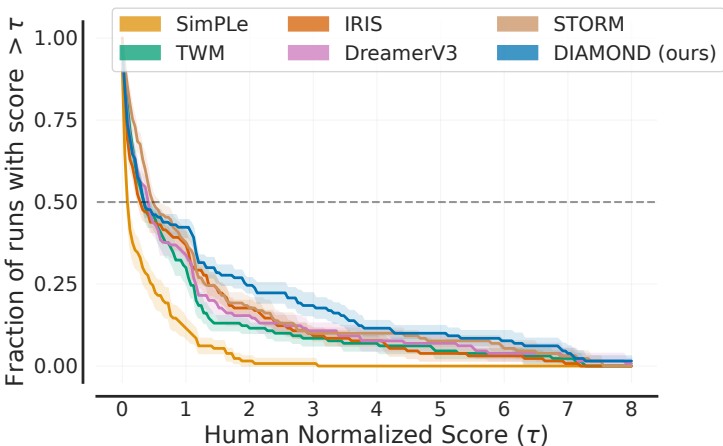

Figure 7: Performance profiles, i.e. fraction of runs above a given human normalized score.

As additional angles of comparison, we also provide parameter counts and approximate training times for IRIS, DreamerV3 and DIAMOND in Table 4 below. We see that DIAMOND has the highest mean HNS, with fewer parameters than both IRIS and DreamerV3. DIAMOND also trains faster than IRIS, although is slower than DreamerV3.

Table 4: Number of parameters, training time, and mean human-normalized score (HNS).

|  | IRIS | DreamerV3 | DIAMOND (ours) |
|---|---|---|---|
| #parameters (↓) | 30M | 18M | **13M** |
| Training days (↓) | 4.1 | **<1** | 2.9 |
| Mean HNS (↑) | 1.046 | 1.097 | **1.459** |

A full training time profile for DIAMOND is provided in Appendix I.

# I  Training time profile

Table 5 provides a full training time profile for DIAMOND.

Table 5: Detailed breakdown of training time. Profiling performed using a Nvidia RTX 4090 with the default hyperparameters specified in Appendices D and E These profiling measures are representative, since exact durations will depend on the machine, the environment, and the training stage.

| Single update | Time (ms) | Detail (ms) |
|---|---|---|
| Total | 543 | $88 + 115 + 340$ |
|     Diffusion model update | 88 | - |
|     Reward/Termination model update | 115 | - |
|     Actor-Critic model update | 340 | $15 \times 20.4 + 34$ |
|         Imagination step (x 15) | 20.4 | $12.7 + 7.0 + 0.7$ |
|             Next observation prediction | 12.7 | $3 \times 4.2$ |
|                 Denoising step (x 3) | 4.2 | - |
|             Reward/Termination prediction | 7.0 | - |
|             Action prediction | 0.7 | - |
|         Loss computation and backward | 34 | - |
| **Epoch** | **Time (s)** | **Detail (s)** |
| Total | 217 | $35 + 46 + 136$ |
|     Diffusion model | 35 | $400 \times 88 \times 10^{-3}$ |
|     Reward/Termination model | 46 | $400 \times 115 \times 10^{-3}$ |
|     Actor-Critic model | 136 | $400 \times 340 \times 10^{-3}$ |
| **Run** | **Time (days)** | **Detail (days)** |
| Total | 2.9 | $2.5 + 0.4$ |
|     Training time | 2.5 | $1000 \times 217/(24 \times 3600)$ |
|     Other (collection, evaluation, checkpointing) | 0.4 | - |

# J Broader comparison to model-free and search-based methods

Table 6 provides scores for model-free and search-based methods, including the current best performing methods on the Atari 100k benchmark, EfficientZero (Ye et al., 2021) and BBF (Schwarzer et al., 2023). Both of these methods use approaches that are out of scope of our approach, such as computationally expensive lookahead Monte-Carlo tree search for EfficientZero, and using periodic network resets in combination with hyperparameter scheduling for BBF. We see that while the use of lookahead search and more advanced reinforcement learning techniques (for EfficientZero (Ye et al., 2021) and BBF (Schwarzer et al., 2023) respectively) can still provide greater performance overall, DIAMOND promisingly still outperforms these methods on some games.

Table 6: Raw scores and human-normalized metrics for search-based and model-free methods.

| Game | Human | Search-based | | Model-free | | | | DIAMOND (ours) |
| | | MuZero | EfficientZero | CURL | SPR | SR-SPR | BBF | |
| --- | --- | --- | --- | --- | --- | --- | --- | --- |
| Alien | 7127.7 | 530.0 | 808.5 | 711.0 | 841.9 | 1107.8 | **1173.2** | 744.1 |
| Amidar | 1719.5 | 38.8 | 148.6 | 113.7 | 179.7 | 203.4 | **244.6** | 225.8 |
| Assault | 742.0 | 500.1 | 1263.1 | 500.9 | 565.6 | 1088.9 | **2098.5** | 1526.4 |
| Asterix | 8503.3 | 1734.0 | **25557.8** | 567.2 | 962.5 | 903.1 | 3946.1 | 3698.5 |
| BankHeist | 753.1 | 192.5 | 351.0 | 65.3 | 345.4 | 531.7 | **732.9** | 19.7 |
| BattleZone | 37187.5 | 7687.5 | 13871.2 | 8997.8 | 14834.1 | 17671.0 | **24459.8** | 4702.0 |
| Boxing | 12.1 | 15.1 | 52.7 | 0.9 | 35.7 | 45.8 | 85.8 | **86.9** |
| Breakout | 30.5 | 48.0 | **414.1** | 2.6 | 19.6 | 25.5 | 370.6 | 132.5 |
| ChopperCommand | 7387.8 | 1350.0 | 1117.3 | 783.5 | 946.3 | 2362.1 | **7549.3** | 1369.8 |
| CrazyClimber | 35829.4 | 56937.0 | 83940.2 | 9154.4 | 36700.5 | 45544.1 | 58431.8 | **99167.8** |
| DemonAttack | 1971.0 | 3527.0 | 13003.9 | 646.5 | 517.6 | 2814.4 | **13341.4** | 288.1 |
| Freeway | 29.6 | 21.8 | 21.8 | 28.3 | 19.3 | 25.4 | 25.5 | **33.3** |
| Frostbite | 4334.7 | 255.0 | 296.3 | 1226.5 | 1170.7 | **2584.8** | 2384.8 | 274.1 |
| Gopher | 2412.5 | 1256.0 | 3260.3 | 400.9 | 660.6 | 712.4 | 1331.2 | **5897.9** |
| Hero | 30826.4 | 3095.0 | **9315.9** | 4987.7 | 5858.6 | 8524.0 | 7818.6 | 5621.8 |
| Jamesbond | 302.8 | 87.5 | 517.0 | 331.0 | 366.5 | 389.1 | **1129.6** | 427.4 |
| Kangaroo | 3035.0 | 62.5 | 724.1 | 740.2 | 3617.4 | 3631.7 | **6614.7** | 5382.2 |
| Krull | 2665.5 | 4890.8 | 5663.3 | 3049.2 | 3681.6 | 5911.8 | 8223.4 | **8610.1** |
| KungFuMaster | 22736.3 | 18813.0 | **30944.8** | 8155.6 | 14783.2 | 18649.4 | 18991.7 | 18713.6 |
| MsPacman | 6951.6 | 1265.6 | 1281.2 | 1064.0 | 1318.4 | 1574.1 | **2008.3** | 1958.2 |
| Pong | 14.6 | -6.7 | 20.1 | -18.5 | -5.4 | 2.9 | 16.7 | **20.4** |
| PrivateEye | 69571.3 | 56.3 | 96.7 | 81.9 | 86.0 | 97.9 | 40.5 | **114.3** |
| Qbert | 13455.0 | 3952.0 | **13781.9** | 727.0 | 866.3 | 4044.1 | 4447.1 | 4499.3 |
| RoadRunner | 7845.0 | 2500.0 | 17751.3 | 5006.1 | 12213.1 | 13463.4 | **33426.8** | 20673.2 |
| Seaquest | 42054.7 | 208.0 | 1100.2 | 315.2 | 558.1 | 819.0 | **1232.5** | 551.2 |
| UpNDown | 11693.2 | 2896.9 | 17264.2 | 2646.4 | 10859.2 | **112450.3** | 12101.7 | 3856.3 |
| #Superhuman (↑) | N/A | 5 | **14** | 2 | 6 | 9 | 12 | 11 |
| Mean (↑) | 1.000 | 0.562 | 1.943 | 0.261 | 0.616 | 1.271 | **2.247** | 1.459 |
| IQM (↑) | 1.000 | 0.288 | 1.047 | 0.113 | 0.337 | 0.700 | **1.139** | 0.641 |

## K Quantitative analysis of autoregressive model drift

Figure 8 provides a quantitative measure of the compounding error demonstrated qualitatively in Figure 3 for DDPM and EDM based world models.

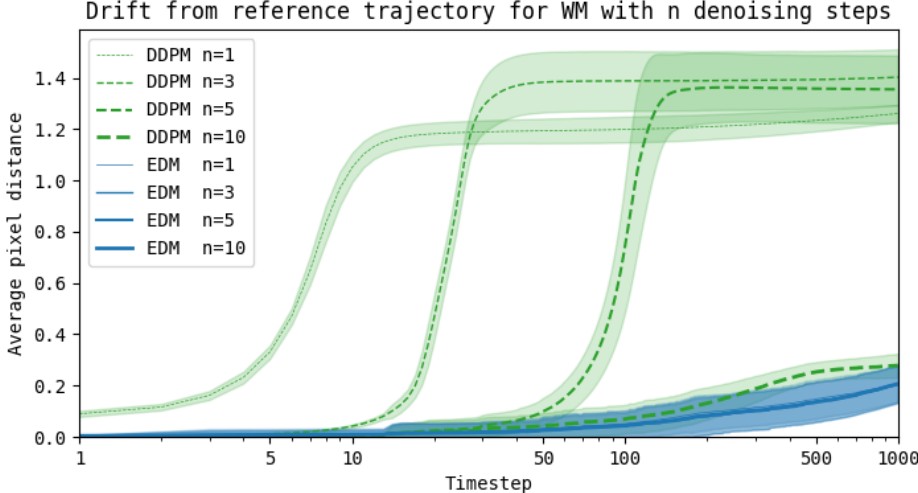

Figure 8: Average pixel drift between an imagined trajectory and the corresponding reference trajectory collected with an expert in *Breakout*. The trajectories are each 1000 timesteps, starting from the same frame and following the same sequence of actions. Each line displays the average and shaded standard deviation of 400 reference trajectories held out from training data. DDPM becomes more stable with increasing number of denoising steps, but is less stable than 1-step EDM, even with 10 denoising steps. The drift we observe for EDM corresponds to differences in the imagined trajectory rather than a pathological color shift as we see in Figure 3a.

## L Quantitative ablation on reducing the number of denoising steps

Table 7 provides a quantitative ablation of the effect of reducing the number of denoising steps used for our EDM diffusion world model from 3 (used for Table 1) to 1, for DIAMOND's 10 highest performing games. Note that the 1-step results correspond to a single seed only so will have higher variance. Nonetheless, these results provide some signal that agents trained with 1 denoising step perform worse than our default choice of 3, particularly for the game *Boxing*, despite the apparent similarity in Figure 8. This additional evidence supports our qualitative analysis in Section 5.2.

Table 7: Quantitative ablation on reducing the number of denoising steps from 3 (default) to 1.

| Game | Random | Human | DIAMOND ($n = 3$) | DIAMOND ($n = 1$) |
|---|---|---|---|---|
| Amidar | 5.8 | 1719.5 | **225.8** | 191.8 |
| Assault | 222.4 | 742.0 | **1526.4** | 782.5 |
| Asterix | 210.0 | 8503.3 | 3698.5 | **6687.0** |
| Boxing | 0.1 | 12.1 | **86.9** | 41.9 |
| Breakout | 1.7 | 30.5 | **132.5** | 50.8 |
| CrazyClimber | 10780.5 | 35829.4 | **99167.8** | 87233.0 |
| Kangaroo | 52.0 | 3035.0 | **5382.2** | 1710.0 |
| Krull | 1598.0 | 2665.5 | 8610.1 | **9105.1** |
| Pong | -20.7 | 14.6 | 20.4 | **20.9** |
| RoadRunner | 11.5 | 7845.0 | **20673.2** | 5084.0 |
| Mean HNS (↑) | 0.000 | 1.000 | **3.052** | 1.962 |

## M Early investigations on visual quality in more complex environments

In the main body of the paper, we evaluated the utility of DIAMOND for the purpose of training RL agents in a world model on the well-established Atari 100k benchmark (Kaiser et al., 2019), and demonstrated DIAMOND's diffusion world model could be applied to model a more complex 3D environment from the game *Counter-Strike: Global Offensive*. In this section, we provide early experiments investigating the effectiveness of DIAMOND's diffusion world model by directly evaluating the visual quality of the trajectories they generate. The two environments we consider are presented in Section M.1 below.

### M.1 Environments

**CS:GO.** We use the *Counter-Strike: Global Offensive* dataset introduced by Pearce and Zhu (2022). Here we use the *Clean* dataset containing 190k frames (3.3 hours) of high-skill human gameplay, captured on the *Dust II* map. This contains observations and actions (mouse and keyboard) captured at 16Hz. We use 150k frames (2.6 hours) for training and 40k frames (0.7 hours) for evaluation. We resize observations to $64\times64$ pixels, and use no augmentation.

**Motorway driving.** We use the dataset from Santana and Hotz (2016)[5], which contains camera and metadata captured from human drivers on US motorways. We select only trajectories captured in daylight, and exclude the first and last 5 minutes of each trajectory (typically traveling to/from a motorway), leaving 4.4 hours of data. We use five trajectories for training (3.6 hours) and two for testing (0.8 hours). We downsample the dataset to 10Hz, resize observations to $64\times64$, and for actions use the (normalized) steering angle and acceleration. During training, we apply data augmentation of shift & scale, contrast, brightness, and saturation, and mirroring.

We note that the purpose of our investigation is to train and evaluate DIAMOND's diffusion model on these static datasets, and that we do not perform reinforcement learning, since there is no standard reinforcement learning protocol for these environments.

### M.2 Diffusion Model Architectures

We consider two potential diffusion model architectures, summarized in Figure 9.

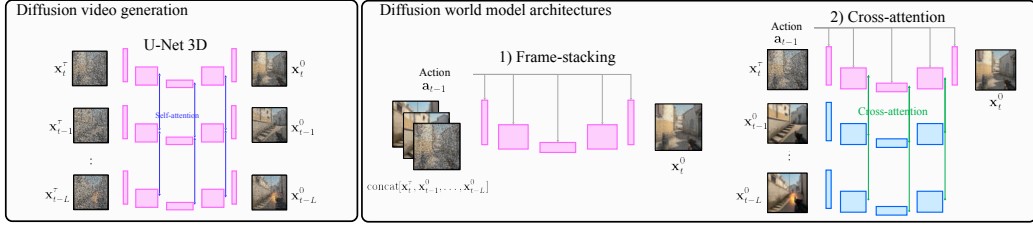

Figure 9: We tested two architectures for DIAMOND's diffusion model which condition on previous image observations in different ways. To illustrate differences with typical video generation models, we also visualize a U-Net 3D (Çiçek et al., 2016) which diffuses a block of frames simultaneously.

**Frame-stacking.** The simplest way to condition on previous observations is by concatenating the previous $L$ frames together with the next noised frame, $\mathrm{concat}[\mathbf{x}_t^\tau, \mathbf{x}_{t-1}^0, \ldots, \mathbf{x}_{t-L}^0]$, which is compatible with a standard U-Net 2D (Ronneberger et al., 2015). This architecture is particularly attractive due to its lightweight construction, requiring minimal additional parameters and compute compared to typical image diffusion. This is the architecture we used for the main body of the paper.

**Cross-attention.** The U-Net 3D (Çiçek et al., 2016), also displayed for comparison in Figure 9, is a leading architecture in video diffusion (Ho et al., 2022). We adapted this design to have an autoregressive cross-attention architecture, formed of a core U-Net 2D, that only receives a single noised frame as direct input, but which cross-attends to the activations of a separate history encoder network. This encoder is a lightweight version of the U-Net 2D architecture. Parameters are shared

---

[5]https://github.com/commaai/research

for all $L$ encoders, and each receives the relative environment timestep embedding as input. The final design differs from the U-Net 3D which diffuses all frames jointly, shares parameters across networks, and uses self-, rather than cross-, attention.

## M.3 Metrics, Baselines and Compute

**Metrics.** To evaluate the visual quality of generated trajectories, we use the standard Fréchet Video Distance (**FVD**) (Unterthiner et al., 2018) as implemented by Skorokhodov et al. (2022). This is computed between 1024 real videos (taken from the test set), and 1024 generated videos, each 16 frames long (1-2 seconds). Models condition on $L = 6$ previous real frames, and the real action sequence. On this same data, we also report the Fréchet Inception Distance (**FID**) (Heusel et al., 2017), which measures the visual quality of individual observations, ignoring the temporal dimension. For these same sets of videos, we also compute the **LPIPS** loss (Zhang et al., 2018) between each *pair* of real/generated observations (Yan et al., 2023). **Sampling rate** describes the number of observations that can be generated, in sequence, by a single Nvidia RTX A6000 GPU, per second.

**Baselines.** We compare against two well-established world model methods; DreamerV3 (Hafner et al., 2023) and IRIS (Micheli et al., 2023), adapting the original implementations to train on a static dataset. We ensured baselines used a similar number of parameters to DIAMOND. Two variants of IRIS are reported; image observations are discretized into $K = 16$ tokens (as used in the original work), or into $K = 64$ tokens (achieved with one less down/up-sampling layer in the autoencoder, see Appendix E of Micheli et al. (2023)), which provide the potential for modeling higher-fidelity visuals.

**Compute.** All models (baselines and DIAMOND) were trained for 120k updates with a batch size of 64, on up to 4×A6000 GPUs. Each training run took between 1-2 days.

## M.4 Analysis

Table 8: Results for 3D environments. These metrics compare observations from real trajectories and generated trajectories. The generated trajectories are conditioned on an initial set of $L = 6$ observations and a real sequence of actions.

| Method | CS:GO | | | Driving | | | Sample rate (Hz) ↑ | Parameters (#) |
|---|---|---|---|---|---|---|---|---|
| | FID ↓ | FVD ↓ | LPIPS ↓ | FID ↓ | FVD ↓ | LPIPS ↓ | | |
| DreamerV3 | 106.8 | 509.1 | 0.173 | 167.5 | 733.7 | 0.160 | 266.7 | 181M |
| IRIS ($K = 16$) | 24.5 | 110.1 | 0.129 | 51.4 | 368.7 | 0.188 | 4.2 | 123M |
| IRIS ($K = 64$) | 22.8 | 85.7 | 0.116 | 44.3 | 276.9 | 0.148 | 1.5 | 111M |
| DIAMOND frame-stack (ours) | 9.6 | 34.8 | 0.107 | 16.7 | 80.3 | 0.058 | 7.4 | 122M |
| DIAMOND cross-attention (ours) | 11.6 | 81.4 | 0.125 | 35.2 | 299.9 | 0.119 | 2.5 | 184M |

Table 8 reports metrics on the visual quality of generated trajectories, along with sampling rates and number of parameters, for the frame-stack and cross-attention DIAMOND architectures, compared to baseline methods. DIAMOND outperforms the baselines across all visual quality metrics. This validates the results seen in the wider video generation literature, where diffusion models currently lead, as discussed in Section 7. The simpler frame-stacking architecture performs better than cross-attention, something surprising given the prevalence of cross-attention in the video generation literature. We believe the inductive bias provided by directly feeding in the input, frame-wise, may be well suited to autoregressive generation. Overall, these results indicate DIAMOND frame-stack > DIAMOND cross-attention ≈ IRIS 64 > IRIS 16 > DreamerV3, which we found corresponds to our intuition from visual inspection.

In terms of sampling rate, DIAMOND frame-stack (with 20 denoising steps) is faster than IRIS ($K = 16$). IRIS suffers from a further 2.8× slow down for the $K = 64$ version, verifying its sample time is bottlenecked by the number of tokens $K$. On the other hand, DreamerV3 is an order of magnitude faster – this derives from its independent, rather than joint, sampling procedure, and the flip-side of this is the low visual quality of its trajectories.

Figure 10 below shows selected examples of the trajectories produced by DIAMOND in CS:GO and motorway driving. The trajectories are plausible, often even at time horizons of reasonable length. In CS:GO, the model accurately generates the correct geometry of the level as it passes through the doorway into a new area of the map. In motorway driving, a car is plausibly imagined overtaking on the left.

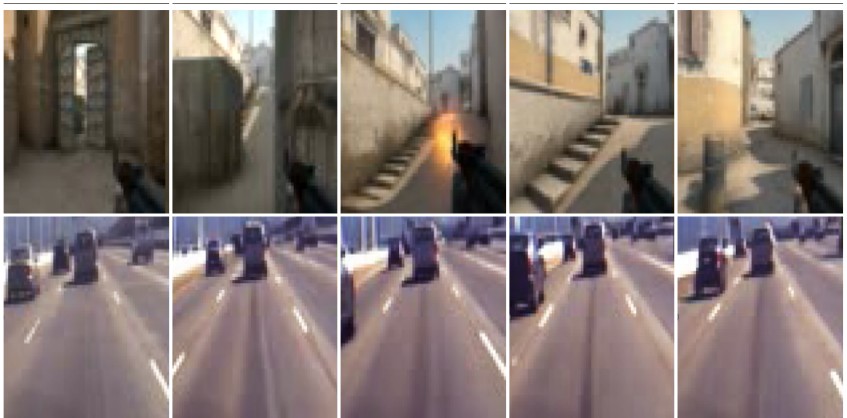

Figure 10: Example trajectories sampled every 25 timesteps from DIAMOND (frame stack) for the modern 3D first-person shooter CS:GO (top row), and real-world motorway driving (bottom row).

While the above experiments use real sequences of actions from the dataset, we also investigated how robust DIAMOND (frame stack) was to novel, user-input actions. Figure 11 shows the effect of the actions in motorway driving – conditioned on the same $L = 6$ real frames, we generate trajectories conditioned on five different action sequences. In general the effects are as intended, e.g. steer straight/left/right moves the camera as expected. Interestingly, when 'slow down' is input, the distance to the car in front decreases since the model predicts that the traffic ahead has come to a standstill. Figure 12 shows similar sequences for CS:GO. For the common actions (mouse movements and fire), the effects are as expected, though they are unstable beyond a few frames, since such a sequence of actions is unlikely to have been seen in the demonstration dataset. We note that these issues – the causal confusion and instabilities – are a symptom of training world models on offline data, rather than being an inherent weakness of DIAMOND.

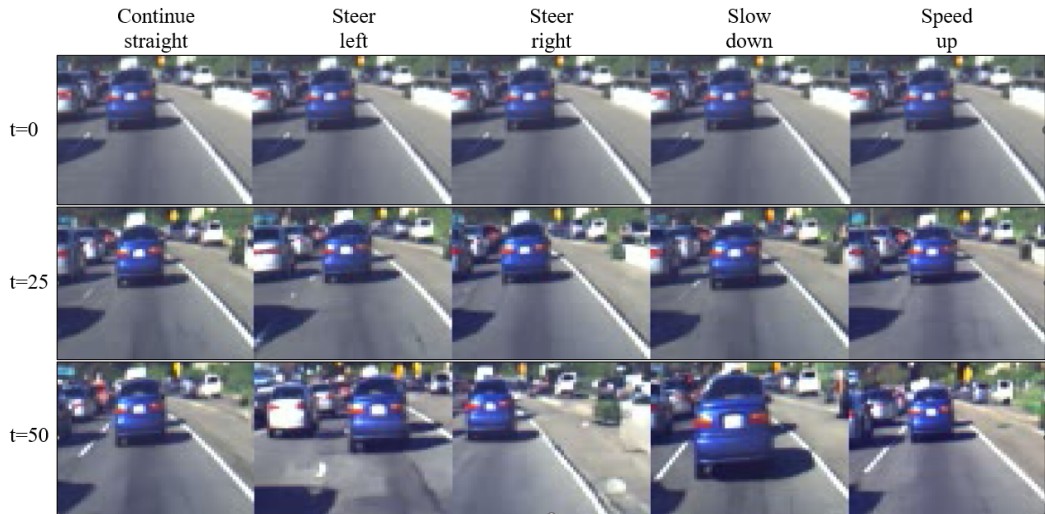

Figure 11: Effect of fixed actions on sampled trajectories in motorway driving. Conditioned on the same initial observations, we rollout the model applying differing actions. Interestingly, the model has learnt to associate 'Slow down' and 'Speed up' actions to the whole traffic slowing down and speeding up.

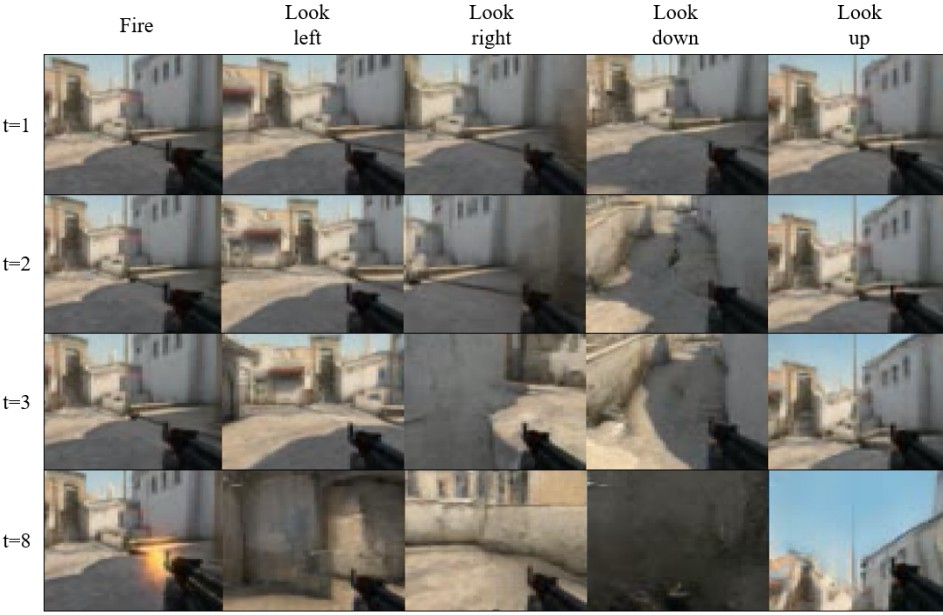

Figure 12: Effect of fixed actions on sampled trajectories in CS:GO. Conditioned on the same initial observation, we rollout the model applying differing actions. Whilst in immediate frames these have the intended effect, for longer roll-outs the observations can degenerate. For instance, it would have been very unlikely for the human demonstrator to look directly into ground in this game state, so the world model is unable to generate a plausible trajectory here, and instead snaps onto another area of the map when looking down does make sense.

