# OpenReview forum: "Diffusion for World Modeling: Visual Details Matter in Atari"
_NeurIPS.cc/2024/Conference — NeurIPS 2024 spotlight_

### Official Review · Reviewer_hqzV · 2024-07-11

**Soundness:** 4
**Presentation:** 4
**Contribution:** 4
**Rating:** 7
**Confidence:** 4

**Summary:**

The paper proposes a visual diffusion-based world model for reinforcement learning. The authors argue that current world models using discrete latent variables may lose important visual details, which are critical for reinforcement learning tasks. DIAMOND addresses this by generating images in the original pixel space. The paper evaluates DIAMOND on the Atari 100k benchmark and achieves strong results relative to other choices for the world model.

**Strengths:**

- Clear and well-written paper
- Strong empirical results beating prior MBRL baselines on Atari 100K
- Impressive scaled-up experiments for visual diffusion world modeling, with consistent visual quality over long horizons even with a low number of diffusion steps
- Good set of ablations over key design choices of the algorithm
- Publicly available code for the experiments

**Weaknesses:**

- It would be useful to include a comparison of training time for DIAMOND compared to prior baselines, e.g. what is the change in speed by using a diffusion world model v.s. Latent world model?
- How does the algorithm compare to the current SOTA model-free algorithms on Atari 100K, e.g. [1]. This would help the author understand the relative utility of MBRL for this setting

Minor:
- Theoretical diffusion background is lengthy and the paper would be better served by including results on other environments that are currently in the appendix, or discussion on architecture used

[1] Bigger, Better, Faster: Human-level Atari with human-level efficiency. Max Schwarzer, Johan Obando-Ceron, Aaron Courville, Marc Bellemare, Rishabh Agarwal, Pablo Samuel Castro.

**Questions:**

See weaknesses.

**Limitations:**

Discussion in paper.

---

> ### Author Rebuttal · Authors · 2024-08-07
>
> Thank you for your clear and concise review. We’re pleased you appreciate our carefully written paper, strong results, ablations and open-source code.
>
> To address your concerns:
>
> > It would be useful to include a comparison of training time for DIAMOND compared to prior baselines, e.g. what is the change in speed by using a diffusion world model v.s. Latent world model?
>
> We agree the training time is an interesting perspective, so had already included a training time comparison with latent world model baselines in Appendix H.
>
> In addition to this comparison with baselines, we have also now included a full training time profile analysis in our additional one-page PDF, which breaks down our total training time in Appendix H into individual steps at the model level. We hope this provides additional insight into the training time of our approach, and we would include this analysis in a camera-ready version of our paper.
>
> > How does the algorithm compare to the current SOTA model-free algorithms on Atari 100K, e.g. [1]. This would help the author understand the relative utility of MBRL for this setting
>
> We also agree that it is helpful to compare to model-free algorithms to understand the relative utility of MBRL, so had already included this comparison to SOTA algorithms such as BBF [1] in Appendix I of our paper. We found it interesting to see that there are some games for which model-free algorithms are dominant and some for which *DIAMOND* performs better.
>
> > Theoretical diffusion background is lengthy and the paper would be better served by including results on other environments that are currently in the appendix, or discussion on architecture used
>
> While we agree that our background on diffusion is more comprehensive than many diffusion papers and that some readers may already have this background, we initially found the diffusion literature to be hard to parse coming from a RL background. Since our target audience is primarily the RL community, we tried our best to provide a concise, pragmatic and clear introduction to diffusion so that our paper can stand alone and be useful to our colleagues in RL. All appendices in the paper, such as the results on other environments and discussions on the architectures used, are referenced from the main text at the relevant points of the paper, so we hope would still be read by the interested reader.
>
> We hope we have addressed all your concerns and questions, and thank you again for your clear and constructive review.

---

> > ### Author Response · Authors · 2024-08-13
> > **Follow up on our rebuttal**
> >
> > Dear Reviewer hqzV,
> >
> > Thank you for your review and for the positive evaluation of our paper. We believe we've addressed your concerns and hope you appreciate our responses and additions.
> >
> > As the discussion period is coming to a close, please let us know if there are any further points we can address to improve your rating.
> >
> > Kind regards,
> >
> > The Authors

---

> > > ### Comment · Reviewer_hqzV · 2024-08-14
> > >
> > > Thank you for the response, I maintain that this is good work that deserves to be published and continue to recommend acceptance!

---

> > > > ### Author Response · Authors · 2024-08-14
> > > >
> > > > Thank you for your response, and thanks again for your positive and constructive review!

---

### Official Review · Reviewer_JScL · 2024-07-11

**Soundness:** 3
**Presentation:** 4
**Contribution:** 3
**Rating:** 7
**Confidence:** 4

**Summary:**

This paper introduces DIAMOND (DIffusion As a Model Of eNvironment Dreams), a novel RL agent trained within a diffusion-based world model. DIAMOND's world model is a diffusion model that, conditioned on past observations and actions, generates an observation at the next time step. This approach diverges from previous methods that rely on discrete latent variables, and offers a promising alternative that leverages the strengths of diffusion models, such as the ability to model multi-modal distributions. The training process is iterative, involving three key steps: data collection in the real environment, world model training on this collected data, and RL agent training with rollouts using the learned world model. The authors analyze design choices for the diffusion world model, and demonstrate that the EDM formulation is superior to DDPM for their use case through visualization of generated trajectories. They also conduct qualitative experiments to determine the optimal number of denoising steps for sampling from the world model. Finally, DIAMOND's evaluation on the Atari 100k benchmark reveals superior performance compared to other world model-based RL agents, achieving a new state-of-the-art mean human-normalized score of 1.46.

**Strengths:**

This paper introduces a new new diffusion-based approach to world modeling in reinforcement learning, and show that it is a promising alternative to prior approaches based on discrete latent variable methods. The proposed method achieves state-of-the-art results on the Atari 100k benchmark and surpasses other world model-based RL agents. The paper also provides a thorough qualitative analysis of the design choices involved in creating an effective diffusion world model, including the choice of diffusion framework (EDM over DDPM) and the number of denoising steps. Overall, the paper is well-written and explains the concepts in a clear manner.

**Weaknesses:**

One minor concern is the lack of quantitative analysis for the design choices. While qualitative experiments and the results on the Atari-100k benchmark effectively motivate the design choices, a more thorough quantitative analysis across the different tasks would provide stronger evidence for these choices. Such a quantitative analysis would be potentially important for applications in real-world environments.

**Questions:**

- The performance on a few tasks (BankHeist, Frostbite, UpNDown) are considerably worse than the baselines. Have you investigated why this is the case?
- Is the diffusion model retrained from scratch in each epoch of the training loop? Also, is the training data in a given epoch just what is collected in the current epoch or the union of data collected in all epochs so far?
- Have you investigated how the diffusion world model evolves as the agent used for collecting the training data improves?

**Limitations:**

The authors have included a dedicated section that discusses the limitations of the work.

---

> ### Author Rebuttal · Authors · 2024-08-07
>
> Thank you for your insightful review. We are pleased that you appreciate the potential of our diffusion-based approach, our thorough analysis of design choices and the clarity of our paper.
>
> We can understand your minor concern with a lack of more quantitative analysis of our design choices. To address this, we have now added a plot displaying the average drift from reference trajectories for DDPM and EDM-based world models for different numbers of denoising steps in our additional one-page PDF. This plot confirms and quantifies the insights provided by our qualitative analysis in Figure 3, and would be included in a camera-ready version of our paper.
>
> While this plot clearly demonstrates the benefits of EDM over DDPM, it does not show any significant difference between the different numbers of denoising steps for EDM. Therefore, we decided to investigate if EDM with 1-step denoising would affect the down-the-line performance of the model-based RL agent, compared to our 3-step default, for our 10 highest performing games. Our results are displayed in Table 1 in our additional one-page PDF. Even though there is some variance due to the fact we only had time to run a single seed for this ablation, we see there is some signal that the agents trained on the 1-step EDM model perform worse, as the mean HNS on these games dropped from 3.1 to 2.0. The drop is particularly evident on the game *Boxing*, which again confirms our qualitative analysis in Figure 4.
>
> These additional quantitative results strengthen the justification for our design choices, and would be included (with additional seeds) in a camera-ready version of our paper.
>
> To address your questions:
>
> > The performance on a few tasks (*BankHeist*, *Frostbite*, *UpNDown*) are considerably worse than the baselines. Have you investigated why this is the case?
>
> While the performance on these games is indeed worse than some baselines, it is generally in-line with the performance of *IRIS*, for which we have a very similar reinforcement learning pipeline, as described in Section 3.2. This suggests that the performance difference is likely due to differences in reinforcement learning (such as hyper parameter choices not being as well suited to these environments) rather than due to differences in the performance of the world model.
>
> > Is the diffusion model retrained from scratch in each epoch of the training loop? Also, is the training data in a given epoch just what is collected in the current epoch or the union of data collected in all epochs so far?
>
> These are indeed important details. The current diffusion model is updated in each epoch, not trained from scratch. The training data used is the union of data collected in all epochs collected so far. These details are mentioned in Section 3.2 and Algorithm 1.
>
> > Have you investigated how the diffusion world model evolves as the agent used for collecting the training data improves?
>
> Yes, we found the world-model is generally quite restricted to the policy of the current agent collecting the training data. In *Breakout* for example, we unsurprisingly found that the world model was not able to predict a brick being broken from a higher row (different color) before this happened in the data collected by the real agent, although it was able to generalize to a brick being broken with a ball coming from a different location.
>
> In any case, we agree that this is useful for the community to be able to investigate, so have added an option `--pick-checkpoint` to select a particular checkpoint to play with to our codebase. In the following code snippet, the training command is modified to save every checkpoint, and the play command includes the new option to pick a checkpoint.
>
> ```bash
> python src/main.py checkpointing.save_agent_every=1 checkpointing.num_to_keep=null
> cd outputs/<DATE>/<TIME>
> python src/play.py --pick-checkpoint
> ```
>
> We hope that our responses have addressed all your questions and believe that our additions following your review have improved our paper, so thank you again for your constructive feedback.

---

> > ### Comment · Reviewer_JScL · 2024-08-13
> > **Thank you for your response**
> >
> > Thank you for the detailed responses to my questions and comments.
> > I have updated my score to reflect this.

---

> > > ### Author Response · Authors · 2024-08-13
> > > **Thank you for your response**
> > >
> > > Thank you for your response and updating your score, and thanks again for your constructive review.

---

### Official Review · Reviewer_h6bf · 2024-07-12

**Soundness:** 3
**Presentation:** 3
**Contribution:** 3
**Rating:** 7
**Confidence:** 3

**Summary:**

This paper introduces a new world model for learning behaviors in imagination using reinforcement learning.
In particular, a diffusion model is used to generate the next frame, conditioned on previous frames and actions.
At each environment step, multiple denoising steps are performed to convert a noise image into the next frame.
By training an RL agent on synthesized trajectories, the method achieves state-of-the-art performance on the Atari 100k benchmark.

**Strengths:**

- S1: The world model achieves strong performance on the Atari 100k benchmark, compared with other world models.
- S2: Implementing world models using diffusion models is a logical step, given the success of diffusion models in image generation.

**Weaknesses:**

- W1: The analysis of the world model is mainly qualitative and not quantitative. The model seems to be really good at generating long trajectories without compounding errors (e.g., Figure 3(b)), but it would be nice to have a more objective measurement of this. One simple idea would be to generate long trajectories and compare the generated frames to the nearest neighbors in the replay buffer. This could also be compared with IRIS and DreamerV3.
- W2: As the training is rather slow, it would be interesting to see a breakdown of the training times of the individual components. For instance, how much time is spent to generate frames compared with the reward/termination model (which is a CNN + LSTM)?

Typo in L240: The single denoising step is shown in the "last row" (instead of "first row").

**Questions:**

- Q1: The imagination horizon is set to the usual value of 15 steps. I am wondering whether longer horizons would lead to better scores, or whether this is not required for Atari?

**Limitations:**

The authors addressed all limitations adequately.

---

> ### Author Rebuttal · Authors · 2024-08-07
>
> Thank you for your clear and concise review. We are pleased that you appreciate our idea and strong results.
>
> Regarding your concern with our analysis being mainly qualitative (W1), we agree that a more quantitative measure of the compounding error of different methods for long trajectories would be valuable. Following your suggestion, we extended our analysis to include a plot demonstrating the average drift in generated observations with respect to reference trajectories (provided in our additional one-page PDF). Specifically, we generated 1000-step trajectories with our world model and the real environment, starting from the same frame and following the same action sequence. This new plot confirms that DDPM suffers from accumulating error for small number of denoising steps, and that EDM is more stable even with low number of denoising steps, as illustrated in Figure 3 of the paper.
>
> While this plot clearly demonstrates the benefits of EDM over DDPM, it does not show any significant difference between the different numbers of denoising steps for EDM. Therefore, we decided to investigate if EDM with 1-step denoising would affect the down-the-line performance of the model-based RL agent, compared to our 3-step default, for our 10 highest performing games. Our results are displayed in Table 1 in our additional one-page PDF. Even though there is some variance due to the fact we only had time to run a single seed for this ablation, we see there is some signal that the agents trained on the 1-step EDM model perform worse, as the mean HNS on these games dropped from 3.1 to 2.0. The drop is particularly evident on the game *Boxing*, which again confirms our qualitative analysis in Figure 4.
>
> These additional quantitative results strengthen the justification for our design choices, and would be included (with additional seeds) in a camera-ready version of our paper.
>
> Regarding the breakdown of training times (W2), following your suggestion we have now included a table in our additional one-page PDF (Table 2) demonstrating the breakdown of our training time into individual model steps. We see that a world model step requires around twice as much time to generate the next frame (12.7ms) compared to the reward/termination prediction (7.0ms) using our default procedure with 3 denoising steps. This breakdown also confirms that integrating the reward prediction into the diffusion model (suggested in our limitation section) would be a promising future direction, as it would likely speed up the world model’s imagination. We hope this provides additional insight into our training time and would include this table in a camera-ready version of our paper.
>
> Thanks for noticing the typo in L240; we have now fixed it.
>
> Regarding your question around the effect of increasing the imagination horizon (Q1), during development we ran some experiments on this question and did not find much signal that increasing the horizon led to better scores on the games we investigated. In any case, increasing the horizon comes at an increased computational cost, since a longer trajectory must be generated for a single agent update. Additionally, this update may be higher variance earlier in training when the world model is less reliable, so we decided to stick with the default value of 15.
>
> We hope we have addressed both of your concerns, and believe your suggestions have improved our paper, so thank you again for your review and constructive feedback!

---

> > ### Comment · Reviewer_h6bf · 2024-08-13
> >
> > Thank you for the thorough response! I have updated my score accordingly.

---

> > > ### Author Response · Authors · 2024-08-13
> > > **Thank you for your response**
> > >
> > > Thank you for your response and updating your score, and thanks again for your constructive review.

---

### Official Review · Reviewer_6rRD · 2024-07-12

**Soundness:** 3
**Presentation:** 3
**Contribution:** 2
**Rating:** 4
**Confidence:** 4

**Summary:**

This paper proposes an approach for learning world models with diffusion based approaches, compared to the recently proposed ones using transformers or the ones dependent on discrete latent variables in general. The core idea is that given success of image generation using diffusion models, the visual details of game playing RL tasks can be improved if a diffusion based world model can be learnt; which can in turn lead to improved performance specifically in pixel based tasks like Atari.

**Strengths:**

The idea of using generative models for learning the world model in RL is perhaps nothing new; past works have tried to do this with transformers or other discrete variable models; however success has not been achieved much. In contrast, while the idea of using diffusion for world models perhaps may sound oblivious given current context, this paper does a good job in demonstrating that this can work empirically in Atari. Experimental comparisons with past works such as IRIS, that are dependent on using transformers, shows that the learnt difufison based approach can achieve empirical gains.

**Weaknesses:**

One primary bottleneck of this approach would be the dependency of the diffusion driven word model for long horizon tasks. Past works often model a per-next time step state prediction dynamics model, or uses models that can handle longer sequences. In contrast, the diffusion based approach may signifcaintly suffer from the long horizon.

Experimental approach and algorithmic novelty is perhaps nothing significantly new; this paper basically does a simple plug in of diffusion model with careful fine-tuning in the context of model based RL.

As stated by the authors as well, the practical choice of the diffusion approach would matter a lot here; and carefully analysis is required to ensure that the difufuson model cxan learn a good world model.

Empirically, other than Atari benchmarks, can this work demonstrate that the idea can work in other domains and compare to more extensive experimental analysis with Dreamer based approaches? For example, Dreamer line of work often shows good results in locomotion or humanoid driven tasks - I think this approach can fail in those cases since it is harder to learn a world model that can generate visual details in those tasks? Does the authors have any comments around that?

**Questions:**

Can we see some more experimental results other than Atari to demonstrate that the idea can work?
More experimental comparisons with other model based approaches are required. MBRL is a huge literature in the field and lots of works have compared to different algorithms in different benchmarks. In contrast, this paper lacks empirical evidence other than Atari domains which seems concerning.
Have the authors considered other non-diffusion based approaches or more recent works based on flow matching for example? Or can we see how the learnt world model differs based on the type of generative model we use? Example, if we can use different variants of diffusion or FM based models, how does the eprfomance differ? I’d like to understand in general, the significance of using generative models for learning world models.
There used to be prior works on imagination augmented rollouts, or works that would learn a multi-step forward dynamics model for example; other than IRIS that are dependent on the use of transformers, can we see some more experimental comparisons with those prior approaches

**Limitations:**

Lack of enough experimental evidence other than Atari benchmarks

---

> ### Author Rebuttal · Authors · 2024-08-07
>
> Thank you for your review. We are pleased you believe our work does a good job in demonstrating that using diffusion for world modeling can work well in Atari and recognize our experimental gains on this competitive benchmark.
>
> Your main concern seems to be the focus of our evaluation on the Atari 100k benchmark. First, many recent works (*IRIS* [1], *TWM* [2], *STORM* [3], *BBF* [4]) were widely adopted having solely evaluated on these 26 Atari 100k games, indicating that they are generally considered to be diverse enough to comprehensively evaluate the advantages and drawbacks of various approaches.
>
> Second, we do apply our diffusion world model to other more visually complex and realistic domains. In particular, we demonstrate that our world model has improved visual quality over *DreamerV3* [5] and *IRIS* on the popular video game *Counter Strike: Global Offensive*, and a real-world motorway driving dataset in Appendix J. We did not consider the less visually complex locomotion tasks to be relevant, but since our method makes no Atari-specific assumptions, our open-source codebase could be applied to other environments of interest by the community.
>
> Regarding comparisons with other non-diffusion model-based approaches you mentioned, we do indeed compare with many non-diffusion methods, including the state-of-the-art model-based approaches on this benchmark, and additional model-free methods in Appendix I. For flow matching approaches, we are not aware of world models that have been designed to leverage this new class of generative models. However, we believe that this would be an interesting direction to investigate in future work, given flow matching provides straighter integration paths more robust to few-step denoising, and enables exact likelihood computation, both of which may be valuable for world modeling.
>
> Thank you again for your insightful review. We hope we have addressed your concern regarding our evaluation, and answered your questions on the place of our work in the broader literature.
>
> ---
>
> **References**
>
> [1] Micheli et al., *Transformers are Sample-Efficient World Models*, ICLR 2023
>
> [2] Robine et al., *Transformer-based World Models Are Happy With 100k Interactions*, ICLR 2023
>
> [3] Zhang et al., *STORM: Efficient Stochastic Transformer based World Models for Reinforcement Learning*, NeurIPS 2023
>
> [4] Schwarzer et al., *Bigger, Better, Faster: Human-level Atari with human-level efficiency*, ICML 2023
>
> [5] Hafner et al., *Mastering diverse domains through world models*, arXiv 2023

---

> ### Author Response · Authors · 2024-08-13
> **Follow up on our rebuttal**
>
> Dear Reviewer 6rRD,
>
> Thank you again for your thorough review of our paper. With less than 24 hours remaining in the discussion period, we would be grateful for your feedback on our response.
>
> Please let us know if there is anything else we can address, as we do feel that your current rating does not fairly reflect the value of our contribution.
>
> Kind regards,
>
> The Authors

---

### Author Rebuttal · Authors · 2024-08-07

We sincerely thank all the reviewers for taking the time to review our paper, and for their positive and constructive feedback.

We are pleased to see a general consensus regarding the motivation of our work, the clarity of our paper, and the strong results achieved by our method.

The main suggestion to improve our paper appeared to be to provide **more quantitative analysis of our ablations** to justify our design decisions. In our paper, we demonstrated visually that an EDM-based diffusion world model is more stable than a DDPM-based world model (Figure 3) and that the use of EDM enabled the use of only 3 denoising steps to provide a fast and reliable world model (Figure 4). **To support this qualitative analysis, we now measure the average drift from reference trajectories** for DDPM and EDM-based world models for different numbers of denoising steps, and provide these results in our additional one-page PDF. **These results confirm and quantify the insights** provided in the qualitative analysis in Figure 3 of the paper.

While this new plot clearly demonstrates the benefits of EDM over DDPM, it does not show any significant difference between the different numbers of denoising steps for EDM. Therefore, we decided to **investigate if EDM with 1-step denoising would affect the downstream performance** of the model-based RL agent, compared to our 3-step default, for our 10 highest performing games. Our results are displayed in Table 1 in our additional one-page PDF. Even though there is some variance due to the fact we only had time to run a single seed for this ablation, we see there is already some signal that the **agents trained on the 1-step EDM model perform worse**, as the mean HNS on these games dropped from 3.1 to 2.0. The drop is particularly evident on the game *Boxing*, which again confirms our qualitative analysis in Figure 4.

**These additional quantitative results strengthen the justification for our design choices**, and would be included (with additional seeds) in a camera-ready version of our paper.

Another point of interest mentioned by multiple reviewers was the **training time of our method**. While we had already included overall training times for comparison with the primary baselines in our paper’s appendix, **we have now included a full profiling analysis** demonstrating the breakdown of our training time into individual model calls in our additional one-page PDF. This breakdown confirms that integrating the reward prediction into the diffusion model (as suggested in our limitation section) would be a promising future direction, as it would likely speed up the world model’s imagination. We would also include this table in a camera-ready version of our paper to **provide additional insight into the training time** and potential improvements to our method.

We hope that we have addressed all of the suggestions raised, and thank all of the reviewers again for their helpful and constructive feedback, which we believe has improved our paper. We look forward to the coming discussion period!

---

> ### Author Response · Authors · 2024-08-11
> **Please could you respond to our rebuttal?**
>
> Dear Reviewers,
>
> Thank you again for your time and effort reviewing our paper. We are now more than halfway through the discussion period. We would be grateful if you could reply to our responses soon so that we have time to address anything remaining.
>
> Please could you let us know if we have addressed your concerns, or if there is anything else we can address for you to consider increasing your ratings of our paper?
>
> Kind regards,
>
> The Authors

---

> > ### Author Response · Authors · 2024-08-12
> > **Follow up on our rebuttal**
> >
> > Dear Reviewers,
> >
> > Thanks again for your time and effort reviewing our work. With less than 48 hours left in the discussion period, we wanted to kindly follow up on our rebuttal. We would greatly appreciate your feedback and are happy to clarify any points if needed.
> >
> > Thank you for your time and consideration.
> >
> > Best regards,
> >
> > The Authors

---

### Decision · Program_Chairs · 2024-09-25

**Decision:**

Accept (spotlight)

**Comment:**

This paper proposes a new RL agent trained with a diffusion-based world model. Authors argue that this approach preserves better visual details than discrete latent models, and the results show SOTA performance on the Atari 100k benchmark .
Authors show strong empirical results on the Atari 100k benchmark, outperforming other world model-based RL agents. This is a novel application of diffusion models to world modeling in RL. The paper is also well written and the code is open sourced.

I would have liked to see more quantitative experiments for the design choices. The method could also be applied to domains beyond Atari. However, reviewers generally agree on the paper's strengths. Three out of four reviewers strongly recommend acceptance, while one reviewer suggests a borderline reject. The authors' responses and additional analyses appear to have satisfied most reviewers.

I recommend accepting this paper. The novel application of diffusion models to world modeling in RL represents an interesting contribution and has potential for several follow up studies.